# Regret Bounds for Risk-Sensitive Reinforcement Learning

**Osbert Bastani**
University of Pennsylvania
obastani@seas.upenn.edu

**Yecheng Jason Ma**
University of Pennsylvania
jasonyma@seas.upenn.edu

**Estelle Shen**
University of Pennsylvania
pixna@sas.upenn.edu

**Wanqiao Xu**
Stanford University
wanqiaox@stanford.edu

## Abstract

In safety-critical applications of reinforcement learning such as healthcare and robotics, it is often desirable to optimize risk-sensitive objectives that account for tail outcomes rather than expected reward. We prove the first regret bounds for reinforcement learning under a general class of risk-sensitive objectives including the popular CVaR objective. Our theory is based on a novel characterization of the CVaR objective as well as a novel optimistic MDP construction.

## 1 Introduction

There has been recent interest in *risk-sensitive reinforcement learning*, which replaces the usual expected reward objective with one that accounts for variation in possible outcomes. One of the most popular risk-sensitive objectives is the *conditional value-at-risk (CVaR)* objective [1, 2, 3, 4], which is the average risk at some tail of the distribution of returns (i.e., cumulative rewards) under a given policy [5, 6]. More generally, we consider a broad class of objectives in the form of a weighted integral of quantiles of the return distribution, of which CVaR is a special case.

A key question is providing regret bounds for risk-sensitive reinforcement learning. While there has been some work studying this question, it has focused on a specific objective called the entropic risk measure [7, 8], leaving open the question of bounds for more general risk-sensitive objectives. There has also been work on optimistic exploration for CVaR [9], but without any regret bounds.

We provide the first regret bounds for risk-sensitive reinforcement learning with objectives of form

$$\Phi(\pi) = \int_0^1 F_{Z^{(\pi)}}^{\dagger}(\tau) \cdot dG(\tau), \tag{1}$$

where $Z^{(\pi)}$ is the random variable encoding the return of policy $\pi$, $F_{Z^{(\pi)}}$ is its quantile function (roughly speaking, the inverse CDF), and $G$ is a weighting function over the quantiles. This class captures a broad range of useful objectives, and has been studied in prior work [10, 4].

We focus on the episodic setting, where the agent interacts with the environment, modeled by a Markov decision process (MDP), over a fixed sequence of episodes. Its goal is to minimize the regret—i.e., the gap between the objective value it achieves compared to the optimal policy. Our approach is based on the upper confidence bound strategy [11, 12], which makes decisions according to an optimistic estimate of the MDP. We prove that this algorithm (denoted $\mathfrak{A}$) has regret

$$\text{regret}(\mathfrak{A}) = \tilde{O}\left(T^2 \cdot L_G \cdot |\mathcal{S}|^{3/2} \cdot |\mathcal{A}| \cdot \sqrt{K}\right),$$

36th Conference on Neural Information Processing Systems (NeurIPS 2022).

where $T$ is the length of a single episode, $L_G$ is the Lipschitz constant for the weighting function $G$, $|\mathcal{S}|$ is the number of states in the MDP, $|\mathcal{A}|$ is the number of actions, and $K$ is the number of episodes (Theorem 4.1). Importantly, it achieves the optimal rate $\sqrt{K}$ achievable for typical expected return objectives (which is a lower bound in our setting since expected return is an objective in the class we consider, taking $G(\tau) = \tau$). For CVaR objectives, we have $L_G = 1/\alpha$, where $\alpha$ is the size of the tail considered—e.g., when $\alpha$ is small, it averages over outliers with particularly small return.

The main challenge behind proving our result is bounding the gap between the objective value for the estimated MDP and the true MDP. In particular, even if we have a uniform bound $\|F_{\hat{Z}^{(\pi)}} - F_{Z^{(\pi)}}\|_\infty$ on the CDFs of the estimated return $\hat{Z}^{(\pi)}$ and the true return $Z^{(\pi)}$, we need to translate this to a bound on the corresponding objective values. To do so, we prove that equivalently, we have

$$\Phi(\pi) = 2T - \int_{\mathbb{R}} G(F_{Z^{(\pi)}}(x)) \cdot dx.$$

This equivalent expression for $\Phi$ follows by variable substitution and integration by parts when $F_{Z^{(\pi)}}$ is invertible (so $F_{Z^{(\pi)}}^{\dagger}(\tau) = F_{Z^{(\pi)}}^{-1}(\tau)$), but the general case requires significantly more care. We show that it holds for an arbitrary CDF $F_{Z^{(\pi)}}$.

In addition to our regret bound, we provide several other useful results for MDPs with risk-sensitive objectives. In particular, optimal policies for risk-sensitive objectives may be non-Markov. For CVaR objectives, it is known that the optimal policy only needs to depend on the cumulative return accrued so far [13]. We prove that this holds in general for objectives of the form (1) (Theorem 3.1). Furthermore, the cumulative return so far is a continuous component; we prove that discretizing this component yields an arbitrarily close approximation of the true MDP (Theorem 3.2).

**Related work.** To the best of our knowledge, the only prior work on regret bounds for risk-sensitive reinforcement learning is specific to the entropic risk objective [7, 8]:

$$J(\pi) = \frac{1}{\beta} \log \mathbb{E}_{Z^{(\pi)}} \left[ e^{\beta Z^{(\pi)}} \right],$$

where $\beta \in \mathbb{R}_{>0}$ is a hyperparameter. As $\beta \to 0$, this objective recovers the expected return objective; for $\beta < 0$, it encourages risk aversion by upweighting negative returns; and for $\beta > 0$, it encourages risk seeking behaviors by upweighting positive returns. This objective is amenable to theoretical analysis since the value function satisfies a variant of the Bellman equation called the *exponential Bellman equation*; however, it is a narrow family of risk measures and is not widely used in practice.

In contrast, we focus on a much broader class of risk measures including the popular CVaR objective [1], which is used to minimize tail losses. To the best of our knowledge, we provide the first regret bounds for the CVaR objective and for the wide range of objectives given by (1).

## 2   Problem Formulation

**Markov decision process.** We consider a Markov decision process (MDP) $\mathcal{M} = (\mathcal{S}, \mathcal{A}, D, P, \mathbb{P}, T)$, with finite state space $\mathcal{S}$, finite action space $\mathcal{A}$, initial state distribution $D(s)$, finite time horizon $T$, transition probabilities $P(s' \mid s, a)$, and reward measure $\mathbb{P}_{R(s,a)}$; without loss of generality, we assume $r \in [0, 1]$ with probability one. A *history* is a sequence

$$\xi \in \mathcal{Z} = \bigcup_{t=1}^{T} \mathcal{Z}_t \qquad \text{where} \qquad \mathcal{Z}_t = (\mathcal{S} \times \mathcal{A} \times \mathbb{R})^{t-1} \times \mathcal{S}$$

Intuitively, a history captures the interaction between an agent and $\mathcal{M}$ up to step $t$. We consider stochastic, time-varying, history-dependent policies $\pi_t(a_t \mid \xi_t)$, where $t$ is the time step. Given $\pi$, the history $\Xi_t^{(\pi)}$ generated by $\pi$ up to step $t$ is a random variable with probability measure

$$\mathbb{P}_{\Xi_t^{(\pi)}}(\xi_t) = \begin{cases} D(s_1) & \text{if } t = 1 \\ \mathbb{P}_{\Xi_{t-1}^{(\pi)}}(\xi_{t-1}) \cdot \pi_t(a_t \mid \xi_{t-1}) \cdot \mathbb{P}_{R(s_t, a_t)}(r_t) \cdot P(s_{t+1} \mid s_t, a_t) & \text{otherwise,} \end{cases}$$

where for all $\tau \in [T]$ we use the notation

$$\xi_\tau = ((s_1, a_1, r_1), ..., (s_{\tau-1}, a_{\tau-1}, r_{\tau-1}), s_\tau).$$

Finally, an *episode* (or *rollout*) is a history $\xi \in \mathcal{Z}_T$ of length $T$ generated by a given policy $\pi$.

**Bellman equation.** The *return* of $\pi$ on step $t$ is the random variable $(Z_t^{(\pi)}(\xi_t))(\xi_T) = \sum_{\tau=t}^{T} r_t$, where $\xi_T \sim \mathbb{P}_{\Xi_T^{(\pi)}}(\cdot \mid \Xi_t^{(\pi)} = \xi_t)$—i.e., it is the reward from step $t$ given that the current history is $\xi_t$. Defining $Z_{T+1}^{(\pi)}(\xi, s) = 0$, the *distributional Bellman equation* [14, 9] is

$$F_{Z_t^{(\pi)}(\xi)}(x) = \sum_{a \in \mathcal{A}} \pi_t(a \mid \xi) \sum_{s' \in \mathcal{S}} P(s' \mid S(\xi), a) \int F_{Z_{t+1}^{(\pi)}(\xi \circ (a, r, s'))}(x - r) \cdot d\mathbb{P}_{R(s,a)}(r),$$

where $S(\xi) = s$ for $\xi = (..., s)$ is the current state in history $\xi$, and $F_X$ is the cumulative distribution function (CDF) of random variable $X$. Finally, the *cumulative return* of $\pi$ is $Z^{(\pi)} = Z_1^{(\pi)}(\xi)$, where $\xi = (s) \in \mathcal{Z}_1$ for $s \sim D$ is the initial history; in particular, we have

$$F_{Z^{(\pi)}}(x) = \int F_{Z_1^{(\pi)}(\xi)}(x) \cdot dD(s).$$

**Risk-sensitive objective.** The *quantile function* of a random variable $X$ is

$$F_X^\dagger(\tau) = \inf \left\{ x \in \mathbb{R} \mid F_X(x) \geq \tau \right\}.$$

Note that if $F_X$ is strictly monotone, then it is invertible and we have $F_X^\dagger(\tau) = F_X^{-1}(\tau)$. Now, our objective is given by the Riemann-Stieljes integral

$$\Phi_{\mathcal{M}}(\pi) = \int_0^1 F_{Z^{(\pi)}}^\dagger(\tau) \cdot dG(\tau),$$

where $G(\tau)$ is a given CDF over quantiles $\tau \in [0, 1]$. This objective was originally studied in [15] for the reinforcement learning setting. For example, choosing $G(\tau) = \min\{\tau/\alpha, 1\}$ (i.e., the CDF of the distribution Uniform$([0, \alpha])$) for $\alpha \in [0, 1]$ yields the $\alpha$-conditional value at risk (CVaR) objective; furthermore, taking $\alpha = 1$ yields the usual expected cumulative reward objective. In addition, choosing $G(\tau) = \mathbb{1}(\tau \leq \alpha)$ for $\alpha \in [0, 1]$ yields the $\alpha$ value at risk (VaR) objective. Other risk sensitive-objectives can also be captured in this form, for example the Wang measure [16], and the cumulative probability weighting (CPW) metric [17]. We call any policy

$$\pi_{\mathcal{M}}^* \in \arg\max_\pi \Phi_{\mathcal{M}}(\pi).$$

an *optimal policy*—i.e., it maximizes the given objective for $\mathcal{M}$.

**Assumptions.** First, we have the following assumption on the quantile function for $Z^{(\pi)}$:

**Assumption 2.1.** $F_{Z^{(\pi)}}^\dagger(1) = T$.

Since $T$ is the maximum reward attainable in an episode, this assumption says that the maximum reward is attained with *some* nontrivial probability. This assumption is very minor; for any given MDP $\mathcal{M}$, we can modify $\mathcal{M}$ to include a path achieving reward $T$ with arbitrarily low probability.

**Assumption 2.2.** $G$ is $L_G$-Lipschitz continuous for some $L_G \in \mathbb{R}_{>0}$, and $G(0) = 0$.

For example, for the $\alpha$-CVaR objective, we have $L_G = 1/\alpha$.

**Assumption 2.3.** We are given an algorithm for computing $\pi_{\mathcal{M}}^*$ for a given MDP $\mathcal{M}$.

For CVaR objectives, existing algorithms [13] can compute $\pi_{\mathcal{M}}^*$ with any desired approximation error. For completeness, we give a formal description of the procedure in Appendix D. When unambiguous, we drop the dependence on $\mathcal{M}$ and simply write $\pi^*$.

Finally, our goal is to learn while interacting with the MDP $\mathcal{M}$ across a fixed number of episodes $K$. In particular, at the beginning of each episode $k \in [K]$, our algorithm chooses a policy $\pi^{(k)} = \mathfrak{A}(H_k)$, where $H_k = \{\xi_{T,\kappa}\}_{\kappa=1}^{k-1}$ is the random set of episodes observed so far, to use for the duration of episode $k$. Then, our goal is to design an algorithm $\mathfrak{A}$ that aims to minimize *regret*, which measures the expected sub-optimality with respect to $\pi^*$:

$$\text{regret}(\mathfrak{A}) = \mathbb{E}\left[ \sum_{k \in [K]} \Phi(\pi^*) - \Phi(\pi^{(k)}) \right].$$

Finally, for simplicity, we assume that the initial state distribution $D$ is known; in practice, we can remove this assumption using a standard strategy.

# 3   Optimal Risk-Sensitive Policies

In this section, we characterize properties of the optimal risk-sensitive policy $\pi^*_{\mathcal{M}}$. First, we show that it suffices to consider policies dependent on the current state and the cumulative rewards obtained so far, rather than the entire history. Second, the cumulative reward is a continuous quantity, making it difficult to compute the optimal policy; we prove that discretizing this component does not significantly reduce the objective value. For CVaR objectives, these results imply that existing algorithms can be used to compute the optimal risk-sensitive policy [13].

**Augmented state space.** We show there exists an optimal policy $\pi^*_t(a_t \mid y_t, s_t)$ that only depends on the current state $s_t$ and cumulative reward $y_t = J(\xi_t) = \sum_{\tau=1}^{t-1} r_\tau$ obtained so far. To this end, let

$$\mathcal{Z}_t(y, s) = \{\xi \in \mathcal{Z}_t \mid J(\xi_t) \leq y \wedge s_t = s\}$$

be the set of length $t$ histories $\xi$ with cumulative reward at most $y$ so far, and current state $s$. For any history-dependent policy $\pi$, define the alternative policy $\tilde{\pi}$ by

$$\tilde{\pi}_t(a_t \mid \xi_t) = \mathbb{E}_{\Xi_t^{(\pi)}} \left[ \pi_t(a_t \mid \Xi_t^{(\pi)}) \,\middle|\, \Xi_t^{(\pi)} \in \mathcal{Z}_t(J(\xi_t), s_t) \right].$$

Note that $\tilde{\pi}$ only depends on $\xi_t$ through $y_t = J(\xi_t)$ and $s_t$, we can define $\tilde{\pi}_t(a_t \mid \xi_t) = \tilde{\pi}_t(a_t \mid y_t, s_t)$.

**Theorem 3.1.** *For any policy $\pi$, we have $\Phi(\tilde{\pi}) = \Phi(\pi)$.*

We give a proof in Appendix A. In particular, given any optimal policy $\pi^*$, we have $\Phi(\tilde{\pi}^*) = \Phi(\pi^*)$; thus, we have $\tilde{\pi}^* \in \arg\max_\pi \Phi(\pi)$. Finally, we note that this result has already been shown for CVaR objectives [13]; our theorem generalizes the existing result to any risk-sensitive objective that can be expressed as a weighted integral of the quantile function.

**Augmented MDP.** As a consequence of Theorem 3.1, it suffices to consider the augmented MDP $\tilde{\mathcal{M}} = (\tilde{\mathcal{S}}, \mathcal{A}, \tilde{D}, \tilde{P}, \tilde{\mathbb{P}}, T)$. First, $\tilde{\mathcal{S}} = \mathcal{S} \times \mathbb{R}$ is the augmented state space; for a state $(s, y) \in \tilde{\mathcal{S}}$, the first component encodes the current state and the second encodes the cumulative rewards so far. The initial state distribution is a probability measure

$$\tilde{D}((s, y)) = D(s) \cdot \delta_0(y),$$

where $\delta_0$ is the Dirac delta measure placing all probability mass on $y = 0$ (i.e., the cumulative reward so far is initially zero). The transitions are given by the product measure

$$\tilde{P}((s', y') \mid (s, y), a) = P(s' \mid s, a) \cdot \mathbb{P}_{R(s,a)}(y' - y),$$

i.e., the second component of the state space is incremented as $y' = y + r$, where $r$ is the reward achieved in the original MDP. Finally, the rewards are now only provided on the final step:

$$\mathbb{P}_{R_t((s,y),a)}(r) = \begin{cases} \delta_y(r) & \text{if } t = T \\ 0 & \text{otherwise}, \end{cases}$$

i.e., the reward at the end of a rollout is simply the cumulative reward so far, as encoded by the second component of the state. By Theorem 3.1, it suffices to compute the optimal policy for $\tilde{\mathcal{M}}$ over history-independent policies $\pi_t(a_t \mid \tilde{s}_t)$:

$$\max_{\pi \in \Pi_{\text{ind}}} \Phi_{\tilde{\mathcal{M}}}(\pi) = \max_\pi \Phi_{\mathcal{M}}(\pi),$$

where $\Pi_{\text{ind}}$ is the set of history-independent policies. Once we have $\pi^*_{\tilde{\mathcal{M}}}$, we can use it in $\mathcal{M}$ by defining $\pi_{\mathcal{M}}(a \mid \xi, s) = \pi^*_{\tilde{\mathcal{M}}}(a \mid J(\xi), s)$.

**Discretized augmented MDP.** Planning over $\tilde{\mathcal{M}}$ is complicated by the fact that the second component of its state space is continuous. Thus, we consider an $\eta$-discretization of $\tilde{\mathcal{M}}$, for some $\eta \in \mathbb{R}_{>0}$. To this end, we modify the reward function so that it only produces rewards in $\eta \cdot \mathbb{N} = \{\eta \cdot n \mid n \in \mathbb{N}\}$, by always rounding the reward up. Then, sums of these rewards are contained in $\eta \cdot \mathbb{N}$, so we can replace the second component of $\tilde{S}$ with $\eta \cdot \mathbb{N}$. In particular, we consider the discretized MDP $\hat{\mathcal{M}} = (\hat{\mathcal{S}}, \mathcal{A}, \tilde{D}, \hat{P}, \tilde{\mathbb{P}}, T)$, where $\hat{\mathcal{S}} = \mathcal{S} \times (\eta \cdot \mathbb{N})$, and transition probability measure

$$\hat{P}((s', y') \mid (s, y), a) = P(s' \mid s, a) \cdot (\mathbb{P}_{R(s,a)} \circ \phi^{-1})(y' - y)$$

where $\phi(r) = \eta \cdot \lceil r/\eta \rceil$. That is, $\mathbb{P}_{R(s,a)}$ is replaced with the pushforward measure $\mathbb{P}_{R(s,a)} \circ \phi^{-1}$, which gives reward $\eta \cdot i$ with probability $\mathbb{P}_{R(s,a)}[\eta \cdot (i-1) < r \leq \eta \cdot i]$.

Now, we prove that the optimal policy $\pi^*_{\hat{\mathcal{M}}}$ for the discretized augmented MDP $\hat{\mathcal{M}}$ achieves objective value close to the optimal policy $\pi^*_{\mathcal{M}}$ for the original MDP $\mathcal{M}$. Importantly, we want to consider measure performance of both policies based on the objective $\Phi_{\mathcal{M}}$ of the original MDP $\mathcal{M}$. To do so, we need a way to use $\pi^*_{\hat{\mathcal{M}}}$ in $\mathcal{M}$. Note that $\pi^*_{\hat{\mathcal{M}}}$ depends only on the state $\hat{s} = (s, y)$, where $s \in \mathcal{S}$ is a state of the original MDP $\mathcal{M}$, and $y \in \eta \cdot \mathbb{N}$ is a discretized version of the cumulative reward obtained so far. Thus, we can run $\pi^*_{\hat{\mathcal{M}}}$ in $\mathcal{M}$ by simply rounding the reward $r_t$ at each step $t$ up to the nearest value $\hat{r}_t \in \eta \cdot \mathbb{N}$ at each step—i.e., $\hat{r}_t = \phi(r_t)$; then, we increment the internal state as $y_t = y_{t-1} + \hat{r}_t$. We call the resulting policy $\pi_{\mathcal{M}}$ the version of $\pi^*_{\hat{\mathcal{M}}}$ *adapted* to $\mathcal{M}$. Then, our next result says that the performance of $\pi_{\mathcal{M}}$ is not too much worse than the performance of $\pi^*_{\mathcal{M}}$.

**Theorem 3.2.** *Let $\pi^*_{\hat{\mathcal{M}}}$ be the optimal policy for the discretized augmented MDP $\hat{\mathcal{M}}$, let $\pi_{\mathcal{M}}$ be the policy $\pi^*_{\hat{\mathcal{M}}}$ adapted to the original MDP $\mathcal{M}$, and let $\pi^*_{\mathcal{M}}$ be the optimal (history-dependent) policy for the original MDP $\mathcal{M}$. Then, we have*

$$\Phi_{\mathcal{M}}(\pi_{\mathcal{M}}) \geq \Phi_{\mathcal{M}}(\pi^*_{\mathcal{M}}) - \eta.$$

We give a proof in Appendix B. Note that we can set $\eta$ to be sufficiently small to achieve any desired error level (i.e., choose $\epsilon/T$, where $\epsilon$ is the desired error). The only cost is in computation time. Note that the number of states in $\hat{\mathcal{M}}$ is still infinite; however, since the cumulative return satisfies $y \in [0, H]$, it suffices to take $\hat{\mathcal{S}} = \mathcal{S} \times (\epsilon \cdot [\lceil H/\eta \rceil])$; then, $\hat{\mathcal{M}}$ has $|\hat{\mathcal{S}}| = |\mathcal{S}| \cdot \lceil H/\eta \rceil$ states.

## 4 Upper Confidence Bound Algorithm

Here, we present our upper confidence bound (UCB) algorithm (summarized in Algorithm 1). At a high level, for each episode, our algorithm constructs an estimate $\mathcal{M}^{(k)}$ of the underlying MDP $\mathcal{M}$ based on the prior episodes $i \in [k-1]$; to ensure exploration, it optimistically inflates the estimate of the reward probability measure $\mathbb{P}$. Then, it plans in $\mathcal{M}^{(k)}$ to obtain an optimistic policy $\pi^{(k)} = \pi^*_{\mathcal{M}^{(k)}}$, and uses this policy to act in the MDP for episode $k$.

**Optimistic MDP.** We define $\mathcal{M}^{(k)}$. Without loss of generality, we assume $\mathcal{S}$ includes a distinguished state $s_\infty$ with rewards $F_{R(s_\infty, a)}(r) = \mathbb{1}(r \geq 1)$ (i.e., achieve the maximum reward $r = 1$ with probability one), and transitions $P(s_\infty \mid s, a) = \mathbb{1}(s = s_\infty)$ and $P(s' \mid s_\infty, a) = \mathbb{1}(s' = s_\infty)$ (i.e., inaccessible from other states and only transitions to itself). Our construction of $\hat{\mathcal{M}}^{(k)}$ uses $s_\infty$ for optimism. Now, let $\tilde{\mathcal{M}}^{(k)}$ be the MDP using the empirical estimates of the transitions and rewards:

$$\tilde{P}^{(k)}(s' \mid s, a) = \frac{N_{k,t}(s, a, s')}{N_{k,t}(s, a)}$$

$$F_{\tilde{R}^{(k)}(s,a)}(r) = \frac{1}{N_{k,t}(s, a)} \sum_{i=1}^{k-1} \sum_{t=1}^{T} \mathbb{1}(r \leq r_{i,t}) \cdot \mathbb{1}(s_{i,t} = s \wedge a_{i,t} = a).$$

Then, let $\hat{\mathcal{M}}^{(k)}$ be the optimistic MDP; in particular, its transitions

$$\hat{P}^{(k)}(s' \mid s, a) = \begin{cases} \mathbb{1}(s' = s_\infty) & \text{if } s = s_\infty \\ 1 - \sum_{s' \in \mathcal{S} \setminus \{s_\infty\}} \tilde{P}^{(k)}(s' \mid s, a) & \text{if } s' = s_\infty \\ \max\left\{ \tilde{P}^{(k)}(s' \mid s, a) - \epsilon_R^{(k)}(s, a),\, 0 \right\} & \text{otherwise} \end{cases}$$

transition to the optimistic state $s_\infty$ when uncertain, and its rewards

$$F_{\hat{R}^{(k)}(s,a)}(r) = \begin{cases} \mathbb{1}(r \geq 1) & \text{if } s = s_\infty \\ 1 & \text{if } r \geq 1 \\ \max\left\{ F_{\tilde{R}^{(k)}(s,a)}(r) - \epsilon_R^{(k)}(s, a),\, 0 \right\} & \text{otherwise} \end{cases}$$

optimistically shift the reward CDF downwards. Here, $\epsilon_P^{(k)}(s, a)$ and $\epsilon_R^{(k)}(s, a)$ are defined in Section 5; intuitively, they are high-probability upper bounds on the errors of the empirical estimates $\tilde{P}^{(k)}(\cdot \mid s, a)$ and $F_{\tilde{R}^{(k)}(s,a)}$ of the transitions and rewards, respectively.

---

**Algorithm 1** Upper Confidence Bound Algorithm

---
1: **for** $k \in [K]$ **do**
2:     Compute $\mathcal{M}^{(k)}$ and $\pi^{(k)} = \pi^*_{\mathcal{M}^{(k)}}$ using prior episodes $\{\xi^{(i)} \mid i \in [k-1]\}$
3:     Execute $\pi^{(k)}$ in the true MDP $\mathcal{M}$ and observe episode $\xi^{(k)} = [(s_{k,t}, a_{k,t}, r_{k,t})]_{t=1}^{T} \cup [s_{k,T+1}]$
4: **end for**

---

**Theoretical guarantees.** We have the following upper bound on the regret of Algorithm 1.

**Theorem 4.1.** *Denote Algorithm 1 by $\mathfrak{A}$. For any $\delta \in (0,1]$, with probability at least $1 - \delta$, we have*

$$\text{regret}(\mathfrak{A}) \leq 4T^{3/2} \cdot L_G \cdot |\mathcal{S}| \cdot \sqrt{5|\mathcal{S}| \cdot |\mathcal{A}| \cdot K \cdot \log\left(\frac{4|\mathcal{S}| \cdot |\mathcal{A}| \cdot K}{\delta}\right)} = \tilde{\mathcal{O}}(\sqrt{K}).$$

We briefly compare our bound to existing ones in the setting of expected return objectives. The dependence on the number of episodes $K$ matches existing bounds [11, 12]; since this is optimal in the setting of expected return, and expected return is a special case of our setting (with $G(\tau) = \tau$), our bound is also optimal in $K$. In terms of the dependence on the number of states $|\mathcal{S}|$, our bound has an extra $\sqrt{|\mathcal{S}|}$ factor compared to the UCRL2 algorithm [11], and an extra $|\mathcal{S}|$ factor compared to the improved bound of the UCBVI algorithm [12]. One extra $\sqrt{|\mathcal{S}|}$ comes from down-shifting transitions uniformly in the construction of the optimistic MDP $\hat{\mathcal{M}}^{(k)}$. This $\sqrt{|\mathcal{S}|}$ may be removed by a more careful construction of the optimistic MDP. Another extra $\sqrt{|\mathcal{S}|}$ compared to UCBVI comes from bounding the estimation error of the reward distribution. We believe it may be possible to remove this $\sqrt{|\mathcal{S}|}$ through a more careful treatment of the estimation error, similar to the one in UCBVI. We leave both of these potential refinements to future work.

In terms of the dependence on the number of actions $|\mathcal{A}|$, our bound matches the order of $\sqrt{|\mathcal{A}|}$ in both UCRL2 and UCBVI. Our dependence on the horizon length $T$ is $T^{3/2}$, compared to the same order of $T^{3/2}$ in UCBVI and $T$ in a variant of UCBVI [12] utilizing a carefully designed variance-based bonus.

## 5 Proof of Theorem 4.1

We prove Theorem 4.1; we defer proofs of several lemmas to Appendix C.

At a high level, the proof proceeds in three steps. First, we prove our key Lemma 5.1, which expresses the objective $\Phi$ in terms of an integral of the weighted CDF of the return. This lemma allows us to translate bounds on the difference between CDFs of the estimated return $\hat{Z}^{(\pi)}$ and the true return $Z^{(\pi)}$ into bounds on the difference between corresponding objective values. The proof of this lemma is divided into three parts that deal with different sets of points in the domain of the quantile function $F^{\dagger}_{Z^{(\pi)}}$: (i) discontinuous; (ii) continuous and strictly monotone; (iii) continuous and non-strictly monotone. This result is used throughout the remainder of the proof.

Second, we define $\mathcal{E}$ to be the event where the optimistic estimated MDP $\hat{\mathcal{M}}^{(k)}$ falls into a certain confidence set around the true MDP $\mathcal{M}$ for each $k \in [K]$; in Lemma 5.2, we prove that $\mathcal{E}$ holds with high probability. Then, in Lemma 5.6, we prove that under event $\mathcal{E}$, the objective values of $\hat{\mathcal{M}}^{(k)}$ and $\mathcal{M}$ are close. To prove this lemma, we separately show that (i) the objective values of the estimated MDP $\tilde{\mathcal{M}}^{(k)}$ (estimated without optimism) and $\mathcal{M}$ are close (Lemma 5.4), and (ii) the objective values of $\hat{\mathcal{M}}^{(k)}$ and $\tilde{\mathcal{M}}^{(k)}$ are close (Lemma 5.5).

Third, in Lemma 5.7, we prove that under event $\mathcal{E}$, the MDP $\hat{\mathcal{M}}^{(k)}$ is indeed optimistic. Together, these results imply the regret bound using the standard UCB proof strategy.

We proceed with the proof. First, we have our key result providing an equivalent expression for $\Phi$:

**Lemma 5.1.** *We have*

$$\Phi(\pi) = T - \int_{\mathbb{R}} G(F_{Z^{(\pi)}}(x)) \cdot dx.$$

*Proof.* First, note that by integration by parts, we have

$$\Phi(\pi) = \int_0^1 F^\dagger_{Z(\pi)}(\tau) \cdot dG(\tau) = \left[ F^\dagger_{Z(\pi)}(\tau) \cdot G(\tau) \right]_0^1 - \int_0^1 G(\tau) \cdot dF^\dagger_{Z(\pi)}(\tau)$$

$$= T - \int_0^1 G(\tau) \cdot dF^\dagger_{Z(\pi)}(\tau),$$

where the last line follows by Assumptions 2.1 & 2.2. Thus, it suffices to show that

$$\int_0^1 G(\tau) \cdot dF^\dagger_{Z(\pi)}(\tau) = \int_\mathbb{R} G(F_{Z(\pi)}(x)) \cdot dx.$$

The quantile function $F^\dagger_{Z(\pi)}$ is monotonically increasing and left-continuous [18], so this integral is equivalently a Lebesgue-Stieltjes integral [19]. Dividing the unit interval $I = [0, 1]$ into disjoint sets

$$I^{(1)} = \{\tau \in \mathbb{I} \mid F^\dagger_{Z(\pi)}(\tau) \text{ is discontinuous}\}$$
$$I^{(2)} = \{\tau \in \mathbb{I} \mid F^\dagger_{Z(\pi)}(\tau) \text{ is continuous and strictly monotone}\}$$
$$I^{(3)} = \{\tau \in \mathbb{I} \mid F^\dagger_{Z(\pi)}(\tau) \text{ is continuous and non-strictly monotone}\},$$

then we have

$$\int_0^1 G(\tau) \cdot dF^\dagger_{Z(\pi)}(\tau) = \int_{I^{(1)}} G(\tau) \cdot dF^\dagger_{Z(\pi)}(\tau) + \int_{I^{(2)}} G(\tau) \cdot dF^\dagger_{Z(\pi)}(\tau) + \int_{I^{(3)}} G(\tau) \cdot dF^\dagger_{Z(\pi)}(\tau).$$

We consider each of the three terms separately and then combine them to finish the proof.

**First term.** Note that $I^{(1)} = \{\tau_i^{(1)}\}_{i=1}^\infty$ is countable since monotone functions can have countably many discontinuities. Also, for each $i \in \mathbb{N}$, the measure assigned to $\tau_i^{(1)}$ by $dF^\dagger_{Z(\pi)}$ is

$$dF^\dagger_{Z(\pi)}(\{\tau_i^{(1)}\}) = \lim_{\tau \to \tau_i^{(1)}+} F^\dagger_{Z(\pi)}(\tau) - F^\dagger_{Z(\pi)}(\tau_i^{(1)}) =: x_i^{(1)+} - x_i^{(1)}.$$

Thus, we have

$$\int_{I^{(1)}} G(\tau) \cdot dF^\dagger_{Z(\pi)}(\tau) = \sum_{i=1}^\infty G(\tau_i^{(1)}) \cdot (x_i^{(1)+} - x_i^{(1)}) = \sum_{i=1}^\infty G(\tau_i^{(1)}) \cdot \int_{x_i^{(1)}}^{x_i^{(1)+}} dx$$

$$= \sum_{i=1}^\infty \int_{x_i^{(1)}}^{x_i^{(1)+}} G(F_{Z(\pi)}(x)) \cdot dx$$

$$= \sum_{i=1}^\infty \int_{F^{-1}_{Z(\pi)}(\{\tau_i^{(1)}\})} G(F_{Z(\pi)}(x)) \cdot dx$$

$$= \int_{F^{-1}_{Z(\pi)}(I^{(1)})} G(F_{Z(\pi)}(x)) \cdot dx.$$

On the second line, we have used the fact that $F_{Z(\pi)}(x) = \tau_i^{(1)}$ for all $x \in [x_i^{(1)}, x_i^{(1)+})$. To see this fact, note that since $x \geq x_i^{(1)}$, by monotonicity of $F_{Z(\pi)}$, we have $F_{Z(\pi)}(x) \geq F_{Z(\pi)}(x_i^{(1)}) = \tau_i^{(1)}$. Furthermore, if $F_{Z(\pi)}(x) > \tau_i^{(1)}$, then we would have

$$x_i^{(1)+} = \lim_{\tau \to \tau_i^{(1)}+} F^\dagger_{Z(\pi)}(\tau) = \lim_{\tau \to \tau_i^{(1)}+} \inf\{x' \in \mathbb{R} \mid F_{Z(\pi)}(x') \geq \tau\}$$

$$\leq \inf\{x' \in \mathbb{R} \mid F_{Z(\pi)}(x') \geq F_{Z(\pi)}(x)\}$$

$$\leq x,$$

where the first inequality follows since $F_{Z(\pi)}(x) \geq \tau$ for $\tau$ sufficiently close to $\tau_i^{(1)}$, and the second since $x \in \{x' \in \mathbb{R} \mid F_{Z(\pi)}(x') \geq F_{Z(\pi)}(x)\}$. Since we have assumed $x < x_i^{(1)+}$, we have a

contradiction, so $F_{Z^{(\pi)}}(x) \le \tau_i^{(1)}$. Thus, it follows that $F_{Z^{(\pi)}}(x) = \tau_i^{(1)}$, as claimed. The third line follows since

$$F_{Z^{(\pi)}}^{-1}(\{\tau_i^{(1)}\}) = [x_i^{(1)}, x_i^{(1)+}) \qquad \text{or} \qquad F_{Z^{(\pi)}}^{-1}(\{\tau_i^{(1)}\}) = [x_i^{(1)}, x_i^{(1)+}].$$

In particular, for any $x \in F_{Z^{(\pi)}}^{-1}(\{\tau_i^{(1)}\})$, we have $F_{Z^{(\pi)}}(x) = \tau_i^{(1)}$, so

$$x \ge \inf\{x \in \mathbb{R} \mid F_{Z^{(\pi)}}(x) \ge \tau_i^{(1)}\} = x_i^{(1)}.$$

Conversely, we have

$$x_i^{(1)+} = \lim_{\tau \to \tau_i^{(1)}+} F_{Z^{(\pi)}}^{\dagger}(\tau) = \lim_{\tau \to \tau_i^{(1)}+} \inf\{x' \in \mathbb{R} \mid F_{Z^{(\pi)}}(x') \ge \tau\} \ge x$$

since $F_{Z^{(\pi)}}(x') \le \tau_i^{(1)} < \tau$ for all $x' \le x$ so the infimum must be $\ge x$. These two arguments show that $F_{Z^{(\pi)}}^{-1}(\{\tau_i^{(1)}\}) \subseteq [x_i^{(1)}, x_i^{(1)+}]$. The fact that $[x_i^{(1)}, x_i^{(1)+}) \subseteq F_{Z^{(\pi)}}^{-1}(\{\tau_i^{(1)}\})$ follows by the same argument as for the second line. The claim follows. Finally, the fourth line follows since the sets $F_{Z^{(\pi)}}^{-1}(\{\tau_i^{(1)}\})$ are disjoint.

**Second term.** For any $\tau \in I^{(2)}$, then $F_{Z^{(\pi)}}^{-1}$ exists at $\tau$, and we have $F_{Z^{(\pi)}}^{\dagger}(\tau) = F_{Z^{(\pi)}}^{-1}(\tau)$. Thus, by a substitution $\tau = F_{Z^{(\pi)}}(x)$, we have

$$\int_{I^{(2)}} G(\tau) \cdot dF_{Z^{(\pi)}}^{\dagger}(\tau) = \int_{F_{Z^{(\pi)}}^{-1}(I^{(2)})} G(F_{Z^{(\pi)}}(x)) \cdot dx.$$

**Third term.** We can divide $I^{(3)}$ into a union of disjoint intervals $I^{(3)} = \bigcup_{i=1}^{\infty} I_i^{(3)}$, where

$$I_i^{(3)} = \{\tau \in [0,1] \mid F_{Z^{(\pi)}}^{\dagger}(\tau) = x_i^{(3)}\}$$

for some $x_i^{(3)} \in \mathbb{R}$; there are only be countably many such intervals (since each one contains a distinct rational number). Then, we have

$$\int_{I^{(3)}} G(\tau) \cdot dF_{Z^{(\pi)}}^{\dagger}(\tau) = 0 = \int_{F_{Z^{(\pi)}}^{-1}(I^{(3)})} G(F_{Z^{(\pi)}}(x)) \cdot dx,$$

since $F_{Z^{(\pi)}}^{-1}(I^{(3)}) = \{x_i^{(3)}\}_{i=1}^{\infty}$ has measure zero according to the Lebesgue measure $dx$.

**Final proof.** Finally, note that $F_{Z^{(\pi)}}^{-1}(I^{(1)})$, $F_{Z^{(\pi)}}^{-1}(I^{(2)})$, and $F_{Z^{(\pi)}}^{-1}(I^{(3)})$ cover $\mathbb{R}$ and are disjoint except possibly on a set of measure zero, so

$$\int_{F_{Z^{(\pi)}}^{-1}(I^{(1)})} G(F_{Z^{(\pi)}}(x)) \cdot dx + \int_{F_{Z^{(\pi)}}^{-1}(I^{(2)})} G(F_{Z^{(\pi)}}(x)) \cdot dx + \int_{F_{Z^{(\pi)}}^{-1}(I^{(3)})} G(F_{Z^{(\pi)}}(x)) \cdot dx$$

$$= \int_{\mathbb{R}} G(F_{Z^{(\pi)}}(x)) \cdot dx.$$

The claim follows. $\qquad\square$

Next, given $\delta \in \mathbb{R}_{>0}$, define $\mathcal{E}$ to be the event where the following hold:

$$\|\tilde{P}^{(k)}(\cdot \mid s, a) - P(\cdot \mid s, a)\|_1 \le \sqrt{\frac{2|\mathcal{S}|}{N^{(k)}(s,a)} \log\left(\frac{6|\mathcal{S}| \cdot |\mathcal{A}| \cdot K}{\delta}\right)} =: \epsilon_P^{(k)}(s,a) \quad (\forall s \in \mathcal{S}, a \in \mathcal{A})$$

$$\|F_{\tilde{R}^{(k)}(s,a)} - F_{R(s,a)}\|_\infty \le \sqrt{\frac{1}{2N^{(k)}(s,a)} \log\left(\frac{6|\mathcal{S}| \cdot |\mathcal{A}| \cdot K}{\delta}\right)} =: \epsilon_R^{(k)}(s,a) \quad (\forall s \in \mathcal{S}, a \in \mathcal{A})$$

$$\|\tilde{P}^{(k)}(\cdot \mid s, a) - P(\cdot \mid s, a)\|_\infty \le \sqrt{\frac{1}{2N^{(k)}(s,a)} \log\left(\frac{6|\mathcal{S}| \cdot |\mathcal{A}| \cdot K}{\delta}\right)} = \epsilon_R^{(k)}(s,a) \quad (\forall s \in \mathcal{S}, a \in \mathcal{A}).$$

**Lemma 5.2.** *We have* $\mathbb{P}[\mathcal{E} \mid \{N^{(k)}(s,a)\}_{k \in [K], s \in \mathcal{S}, a \in \mathcal{A}}] \ge 1 - \delta$.

Next, let $\tilde{Z}^{(k,\pi)}$ and $\hat{Z}^{(k,\pi)}$ be the returns for policy $\pi$ for $\tilde{\mathcal{M}}^{(k)}$ and $\hat{\mathcal{M}}^{(k)}$, respectively, let $\Phi = \Phi_{\mathcal{M}}$, $\tilde{\Phi}^{(k)} = \Phi_{\tilde{\mathcal{M}}^{(k)}}$, and $\hat{\Phi}^{(k)} = \Phi_{\hat{\mathcal{M}}^{(k)}}$, and let $\pi^* = \pi^*_{\mathcal{M}}$, $\tilde{\pi}^{(k)} = \pi^*_{\tilde{\mathcal{M}}^{(k)}}$, and $\hat{\pi}^{(k)} = \pi^*_{\hat{\mathcal{M}}^{(k)}}$. Now, we prove two key results: (i) $\hat{\Phi}^{(k)}$ is close to $\Phi$, and (ii) $\hat{\Phi}^{(k)}$ is optimistic compared to $\Phi$. To this end, we have the following key lemma; its proof depends critically on Lemma 5.1.

**Lemma 5.3.** *Consider MDPs $\mathcal{M} = (\mathcal{S}, \mathcal{A}, D, P, \mathbb{P}, T)$ and $\mathcal{M}' = (\mathcal{S}, \mathcal{A}, D, P', \mathbb{P}', T)$, such that $\|P'(\cdot \mid s, a) - P(\cdot \mid s, a)\|_1 \le \epsilon_P(s, a)$ and $\|F_{R'(s,a)} - F_{R(s,a)}\|_\infty \le \epsilon_R(s, a)$. Then, we have*

$$|\Phi'(\pi) - \Phi(\pi)| \le T \cdot L_G \cdot B(\pi) \qquad (\forall k \in [K], \pi),$$

*where*

$$B(\pi) = \mathbb{E}_{\Xi_T^{(\pi)}}\left[\sum_{t=1}^T \epsilon_P(s_t, a_t) + \epsilon_R(s_t, a_t)\right].$$

Our next lemma characterizes the connection between $\tilde{\Phi}^{(k)}$ and $\Phi$.

**Lemma 5.4.** *On event $\mathcal{E}$ and conditioned on $\{N^{(k)}(s, a)\}_{k \in [K], s \in \mathcal{S}, a \in \mathcal{A}}$, we have*

$$|\tilde{\Phi}^{(k)}(\pi) - \Phi(\pi)| \le T \cdot L_G \cdot B^{(k)}(\pi) \qquad (\forall k \in [K], \pi),$$

*where*

$$B^{(k)}(\pi) = \mathbb{E}_{\Xi_T^{(\pi)}}\left[\sum_{t=1}^T \epsilon_P^{(k)}(s_t, a_t) + \epsilon_R^{(k)}(s_t, a_t) \,\Big|\, \{N^{(k)}(s, a)\}_{s \in \mathcal{S}, a \in \mathcal{A}}\right].$$

*Proof.* The result follows since on event $\mathcal{E}$ and conditioned on $\{N^{(k)}(s, a)\}_{k \in [K], s \in \mathcal{S}, a \in \mathcal{A}}$, $\tilde{\mathcal{M}}^{(k)}$ and $\mathcal{M}$ satisfy the conditions of Lemma 5.3 for all $k \in [K]$. $\qquad\square$

Our next lemma characterizes the connection between $\hat{\Phi}^{(k)}$ and $\tilde{\Phi}^{(k)}$.

**Lemma 5.5.** *For each $k \in [K]$ and any policy $\pi$, we have*

$$|\hat{\Phi}^{(k)}(\pi) - \tilde{\Phi}^{(k)}(\pi)| \le T \cdot L_G \cdot \sqrt{|\mathcal{S}|} \cdot B^{(k)}(\pi).$$

*Proof.* The result follows since by definition of $\hat{\mathcal{M}}^{(k)}$, $\hat{\mathcal{M}}^{(k)}$ and $\tilde{\mathcal{M}}^{(k)}$ satisfy the condition of Lemma 5.3 with $\epsilon_P(s, a) = 2|\mathcal{S}| \cdot \epsilon_R^{(k)}(s, a) \le \sqrt{|\mathcal{S}|} \cdot \epsilon_P^{(k)}(s, a)$ and $\epsilon_R(s, a) = \epsilon_R^{(k)}(s, a)$ for all $k \in [K]$. $\qquad\square$

Now, we prove the first key claim—i.e., $\hat{\Phi}^{(k)}$ is close to $\Phi$.

**Lemma 5.6.** *On event $\mathcal{E}$, for all $k \in [K]$ and any policy $\pi$, we have*

$$|\hat{\Phi}^{(k)}(\pi) - \Phi(\pi)| \le 2T \cdot L_G \cdot \sqrt{|\mathcal{S}|} \cdot B^{(k)}(\pi).$$

*Proof.* Note that

$$|\hat{\Phi}^{(k)}(\pi) - \Phi(\pi)| \le |\hat{\Phi}^{(k)}(\pi) - \tilde{\Phi}^{(k)}(\pi)| + |\tilde{\Phi}^{(k)}(\pi) - \Phi(\pi)| \le 2T \cdot L_G \cdot \sqrt{|\mathcal{S}|} \cdot B^{(k)}(\pi),$$

where the second inequality follows by Lemmas 5.4 & 5.5. $\qquad\square$

Now, we prove the second key claim—i.e., $\hat{\Phi}^{(k)}$ is optimistic compared to $\Phi$.

**Lemma 5.7.** *On event $\mathcal{E}$, we have $\hat{\Phi}^{(k)}(\pi) \ge \Phi(\pi)$ for all $k \in [K]$ and all policies $\pi$.*

With these two key claims, the proof of Theorem 4.1 follows by a standard upper confidence bound argument; we give the proof in Appendix C.4.

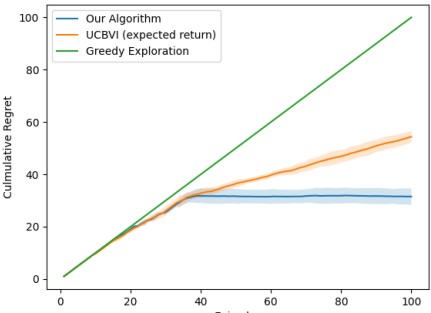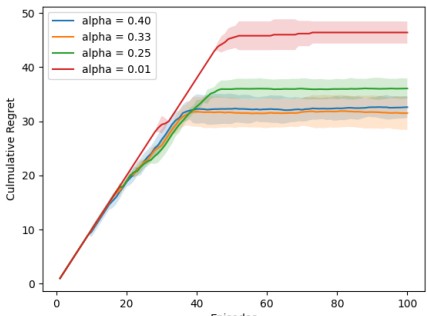

Figure 1: Results on the frozen lake environment. Left: Regret of our algorithm vs. UCBVI (with expected return) and a greedy exploration strategy. Right: Regret of our algorithm across different $\alpha$ values. We show mean and standard deviation across five random seeds.

# 6 Experiments

We consider a classic frozen lake problem with a finite horizon. The agent moves to a block next to its current state at each timestep $t$ and has a slipping probability of 0.1 in its moving direction if the next state is an ice block. The objective is to maximize the cumulative reward without falling into holes. The agent needs to choose among paths which correspond to different levels of risk and rewards. In other words, the agent should account for the tradeoff between the cumulative reward and risk of slipping into holes. We use a map with four paths of the same lengths that have different rewards at the end and different levels of risk of falling into holes. We consider $\alpha \in \{0.40, 0.33, 0.25, 0.01\}$, which correspond to optimal policies of choosing paths with best possible returns of $\{6, 4, 2, 1\}$ and success probabilities of $\{0.729, 0.81, 0.9, 1\}$, respectively (failure corresponds to zero return).

Figure 1 (left) shows the comparison in cumulative regret between our algorithm, UCBVI (which maximizes expected returns, not our risk-sensitive objective), and the an algorithm that optimizes our risk-sensitive objective but explores in a greedy way (i.e., use the best policy for the current estimated MDP without any optimism), for $\alpha = 0.33$. The regret is measured in terms of the CVaR objective with respect to the optimal policy for the same CVaR objective. While UCBVI outperforms greedy, neither of them converge; in contrast, our algorithm converges within 40 episodes.

Figure 1 (right) compares the regret between our algorithm under different values of $\alpha$ using the CVaR objective. Note that smaller values of $\alpha$ tend to lead our algorithm to converge more slowly; this result matches our theory since smaller $\alpha$ corresponds to larger $L_G$. Intuitively, more samples are needed to get a good estimate of the objective as $\alpha$ becomes small since the CVaR objective is the average return over a tiny fraction of samples, causing high variance in our estimate of the objective.

# 7 Conclusion

We have proposed a novel regret bound for risk sensitive reinforcement learning that applies to a broad class of objective functions, including the popular conditional value-at-risk (CVaR) objective. Our results recover the usual $\sqrt{K}$ dependence on the number of episodes, and also highlights dependence on the Lipschitz constant $L_G$ of the integral of the weighting function $G$ used to define the objective. Future work includes extending these ideas to the setting of function approximation and understanding whether alternative exploration strategies such as Thompson sampling are applicable.

## Acknowledgments and Disclosure of Funding

This work is funded in part by NSF Award CCF-1910769, NSF Award CCF-1917852, and ARO Award W911NF-20-1-0080. The U.S. Government is authorized to reproduce and distribute reprints for Government purposes notwithstanding any copyright notation herein.

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
