# A Proof of Theorem 3.1

In this section, we prove Theorem 3.1, which says that it suffices to the augmented state space $(y, s)$ rather than the whole history $\xi$. First, we have the following lemma.

**Lemma A.1.** *For any $y \in \mathbb{R}$ and $s \in \mathcal{S}$, we have*

$$\mathbb{E}_{\Xi_t^{(\pi)}}\left[\pi_t(a \mid \Xi_t^{(\pi)}) \cdot \mathbb{1}\left(\Xi_t^{(\pi)} \in \mathcal{Z}_t(y, s)\right)\right] = \mathbb{E}_{\Xi_t^{(\pi)}}\left[\tilde{\pi}_t(a \mid \Xi_t^{(\pi)}) \cdot \mathbb{1}\left(\Xi_t^{(\pi)} \in \mathcal{Z}_t(y, s)\right)\right].$$

*Proof.* Note that

$$\mathbb{E}_{\Xi_t^{(\pi)}}\left[\tilde{\pi}_t(a \mid \Xi_t^{(\pi)}) \;\middle|\; \Xi_t^{(\pi)} \in \mathcal{Z}_t(y, s)\right]$$

$$= \mathbb{E}_{\Xi_t^{(\pi)}}\left[\mathbb{E}_{\tilde{\Xi}_t^{(\pi)}}\left[\pi_t(a_t \mid \tilde{\Xi}_t^{(\pi)}) \;\middle|\; \tilde{\Xi}_t^{(\pi)} \in \mathcal{Z}_t(J(\Xi_t^{(\pi)}), s_t)\right] \;\middle|\; \Xi_t^{(\pi)} \in \mathcal{Z}_t(y, s)\right]$$

$$= \mathbb{E}_{\Xi_t^{(\pi)}}\left[\mathbb{E}_{\tilde{\Xi}_t^{(\pi)}}\left[\pi_t(a_t \mid \tilde{\Xi}_t^{(\pi)}) \;\middle|\; \tilde{\Xi}_t^{(\pi)} \in \mathcal{Z}_t(y, s)\right] \;\middle|\; \Xi_t^{(\pi)} \in \mathcal{Z}_t(y, s)\right]$$

$$= \mathbb{E}_{\tilde{\Xi}_t^{(\pi)}}\left[\pi_t(a_t \mid \tilde{\Xi}_t^{(\pi)}) \;\middle|\; \Xi_t^{(\pi)} \in \mathcal{Z}_t(y, s)\right].$$

The claim follows by replacing $\tilde{\Xi}_t^{(\pi)}$ with $\Xi_t^{(\pi)}$ and multiplying by $\mathbb{P}_{\Xi_t^{(\pi)}}[\Xi_t^{(\pi)} \in \mathcal{Z}_t(y, s)]$. $\square$

Next, let

$$D_t^{(\pi)}(y, s) = \mathbb{P}_{\Xi_t^{(\pi)}}\left[J(\Xi_t^{(\pi)}) \le y \wedge S(\Xi_t^{(\pi)}) = s\right]$$

be the probability of a history achieving current cumulative return at most $y$ and ending in state $s$.

**Lemma A.2.** *We have $D_t^{(\pi)} = D_t^{(\tilde{\pi})}$.*

*Proof.* We prove by induction. The base case $t = 1$ follows trivially. For the inductive case, note that

$$D_{t+1}^{(\pi)}(y, s'') = \int \mathbb{1}(\xi' \in \mathcal{Z}_{t+1}(y, s'')) \cdot d\mathbb{P}_{\Xi_{t+1}^{(\pi)}}(\xi')$$

$$= \int \sum_{a \in \mathcal{A}} \sum_{s' \in \mathcal{S}} \mathbb{1}(\xi \circ (a, r, s') \in \mathcal{Z}_{t+1}(y, s'')) \cdot P(s' \mid S(\xi), a) \cdot d\mathbb{P}_{R(s, a)}(r) \cdot \pi(a \mid \xi) \cdot d\mathbb{P}_{\Xi_t^{(\pi)}}(\xi)$$

$$= \int \sum_{a \in \mathcal{A}} \sum_{s' \in \mathcal{S}} \mathbb{1}(J(\xi) + r \le y) \cdot \mathbb{1}(s' = s'') \cdot P(s' \mid S(\xi), a) \cdot d\mathbb{P}_{R(s, a)}(r) \cdot \pi(a \mid \xi) \cdot d\mathbb{P}_{\Xi_t^{(\pi)}}(\xi)$$

$$= \int \sum_{a \in \mathcal{A}} \mathbb{1}(J(\xi) + r \le y) \cdot P(s'' \mid S(\xi), a) \cdot d\mathbb{P}_{R(s, a)}(r) \cdot \pi(a \mid \xi) \cdot d\mathbb{P}_{\Xi_t^{(\pi)}}(\xi),$$

where the first line follows by definition of $D_{t+1}^{(\pi)}$, the second by the inductive formula for $\mathbb{P}_{\Xi_{t+1}^{(\pi)}}$, the third since $J(\xi \circ (a, r, s')) = J(\xi) + r$, and the fourth by summing over $s'$. Continuing, we have

$$D_{t+1}^{(\pi)}(y, s'') = \sum_{s \in \mathcal{S}} \sum_{a \in \mathcal{A}} \int \pi(a \mid \xi) \cdot \mathbb{1}(J(\xi) + r \le y) \cdot \mathbb{1}(S(\xi) = s) \cdot d\mathbb{P}_{\Xi_t^{(\pi)}}(\xi) \cdot P(s'' \mid s, a) \cdot d\mathbb{P}_{R(s, a)}(r)$$

$$= \sum_{s \in \mathcal{S}} \sum_{a \in \mathcal{A}} \int \pi(a \mid \xi) \cdot \mathbb{1}(\xi \in \mathcal{Z}_t(y - r, s)) \cdot d\mathbb{P}_{\Xi_t^{(\pi)}}(\xi) \cdot P(s'' \mid s, a) \cdot d\mathbb{P}_{R(s, a)}(r)$$

$$= \sum_{s \in \mathcal{S}} \sum_{a \in \mathcal{A}} \int \tilde{\pi}(a \mid \xi) \cdot \mathbb{1}(\xi \in \mathcal{Z}_t(y - r, s)) \cdot d\mathbb{P}_{\Xi_t^{(\pi)}}(\xi) \cdot P(s'' \mid s, a) \cdot d\mathbb{P}_{R(s, a)}(r),$$

where the first line follows by introducing $\mathbb{1}(S(\xi) = s)$ and rearranging, the second by definition of $\mathcal{Z}_t$, and the third by Lemma A.1. Continuing, we have

$$D_{t+1}^{(\pi)}(y, s'') = \sum_{s \in \mathcal{S}} \sum_{a \in \mathcal{A}} \int \mathbb{1}(\xi \in \mathcal{Z}_t(y - r, s)) \cdot d\mathbb{P}_{\Xi_t^{(\pi)}}(\xi) \cdot \tilde{\pi}(a \mid y - r, s) \cdot P(s'' \mid s, a) \cdot d\mathbb{P}_{R(s,a)}(r)$$

$$= \sum_{s \in \mathcal{S}} \sum_{a \in \mathcal{A}} \int D_t^{(\pi)}(y - r, s) \cdot \tilde{\pi}(a \mid y - r, s) \cdot P(s'' \mid s, a) \cdot d\mathbb{P}_{R(s,a)}(r)$$

$$= \sum_{s \in \mathcal{S}} \sum_{a \in \mathcal{A}} \int D_t^{(\tilde{\pi})}(y - r, s) \cdot \tilde{\pi}(a \mid y - r, s) \cdot P(s'' \mid s, a) \cdot d\mathbb{P}_{R(s,a)}(r)$$

$$= \sum_{s \in \mathcal{S}} \sum_{a \in \mathcal{A}} \int \mathbb{1}(\xi \in \mathcal{Z}_t(y - r, s)) \cdot d\mathbb{P}_{\Xi_t^{(\tilde{\pi})}}(\xi) \cdot \tilde{\pi}(a \mid y - r, s) \cdot P(s'' \mid s, a) \cdot d\mathbb{P}_{R(s,a)}(r),$$

where the first line follows since $\tilde{\pi}$ is independent of $\xi$ and by rearranging, the second by definition of $D_t^{(\pi)}$, the third by induction, and the fourth by definition of $D_t^{(\tilde{\pi})}$. Continuing, we have

$$D_{t+1}^{(\pi)}(y, s'') = \sum_{s \in \mathcal{S}} \sum_{a \in \mathcal{A}} \int \tilde{\pi}(a \mid \xi) \cdot \mathbb{1}(\xi \in \mathcal{Z}_t(y - r, s)) \cdot d\mathbb{P}_{\Xi_t^{(\tilde{\pi})}}(\xi) \cdot P(s'' \mid s, a) \cdot d\mathbb{P}_{R(s,a)}(r)$$

$$= \sum_{s \in \mathcal{S}} \sum_{a \in \mathcal{A}} \int \tilde{\pi}(a \mid \xi) \cdot \mathbb{1}(J(\xi) + r \le y) \cdot \mathbb{1}(S(\xi) = s) \cdot d\mathbb{P}_{\Xi_t^{(\tilde{\pi})}}(\xi) \cdot P(s'' \mid s, a) \cdot d\mathbb{P}_{R(s,a)}(r)$$

$$= \sum_{a \in \mathcal{A}} \int \mathbb{1}(J(\xi) + r \le y) \cdot P(s'' \mid S(\xi), a) \cdot d\mathbb{P}_{R(s,a)}(r) \cdot \tilde{\pi}(a \mid \xi) \cdot d\mathbb{P}_{\Xi_t^{(\tilde{\pi})}}(\xi)$$

$$= \sum_{a \in \mathcal{A}} \sum_{s' \in \mathcal{S}} \int \mathbb{1}(J(\xi) + r \le y) \cdot \mathbb{1}(s' = s'') \cdot P(s' \mid S(\xi), a) \cdot d\mathbb{P}_{R(s,a)}(r) \cdot \tilde{\pi}(a \mid \xi) \cdot d\mathbb{P}_{\Xi_t^{(\tilde{\pi})}}(\xi)$$

$$= \sum_{a \in \mathcal{A}} \sum_{s' \in \mathcal{S}} \int \mathbb{1}(\xi \circ (a, r, s') \in \mathcal{Z}_{t+1}(y, s'')) \cdot P(s' \mid S(\xi), a) \cdot d\mathbb{P}_{R(s,a)}(r) \cdot \tilde{\pi}(a \mid \xi) \cdot d\mathbb{P}_{\Xi_t^{(\tilde{\pi})}}(\xi)$$

$$= \int \mathbb{1}(\xi' \in \mathcal{Z}_{t+1}(y, s'')) \cdot d\mathbb{P}_{\Xi_{t+1}^{(\pi)}}(\xi')$$

$$= D_{t+1}^{(\tilde{\pi})}(y, s''),$$

where the first line follows by definition of $\tilde{\pi}$, the second by definition of $\mathcal{Z}_t$, the third by summing over $s$ and rearranging, the fourth by introducing $\mathbb{1}(s' = s'')$, the fifth by definition of $\mathcal{Z}_{t+1}$, the sixth by the inductive formula for $\mathbb{P}_{\Xi_{t+1}^{(\pi)}}$, and the seventh by the definition of $D_{t+1}^{(\tilde{\pi})}$. The claim follows. □

Now, we prove Theorem 3.1. By Lemma A.2, we have

$$F_{Z^{(\pi)}}(x) = \int \mathbb{1}(J(\xi) \le x) \cdot d\mathbb{P}_{\Xi_T^{(\pi)}}(\xi)$$

$$= \sum_{s \in \mathcal{S}} \int \mathbb{1}(J(\xi) \le x) \cdot \mathbb{1}(S(\xi) = s) \cdot d\mathbb{P}_{\Xi_T^{(\pi)}}(\xi)$$

$$= \sum_{s \in \mathcal{S}} \int \mathbb{1}(\xi \in \mathcal{Z}_T(x, s)) \cdot d\mathbb{P}_{\Xi_T^{(\pi)}}(\xi)$$

$$= \sum_{s \in \mathcal{S}} \int \mathbb{1}(\xi \in \mathcal{Z}_T(x, s)) \cdot d\mathbb{P}_{\Xi_T^{(\tilde{\pi})}}(\xi)$$

$$= \int \mathbb{1}(J(\xi) \le x) \cdot d\mathbb{P}_{\Xi_T^{(\tilde{\pi})}}(\xi)$$

$$= F_{Z^{(\tilde{\pi})}}(x).$$

Theorem 3.1 follows straightforwardly from this result. □

# B  Proof of Theorem 3.2

We construct a sequence of MDPs $\mathcal{M}_0, \mathcal{M}_1, ..., \mathcal{M}_T$, such that $\mathcal{M}_0 = \tilde{\mathcal{M}}$ and $\mathcal{M}_T = \hat{\mathcal{M}}$, and where we can bound the incremental errors

$$\Phi_{\tilde{\mathcal{M}}}(\pi^*_{\mathcal{M}_\tau}) - \Phi_{\hat{M}}(\pi^*_{\mathcal{M}_{\tau-1}}),$$

noting a policy for one of the MDPs can be used in all the other MDPs. For each $\tau \in [T]$, the MDP $\mathcal{M}_\tau$ discretizes the reward assigned on the $t$th step of $\mathcal{M}_{\tau-1}$—more precisely, it discretizes the transitions since the rewards are only assigned on the last step based on the cumulative reward recorded in the second component of the state space. Formally, $\mathcal{M}_\tau$ is identical to $\tilde{\mathcal{M}}$, except it uses the (time-varying) transition probability measure $\hat{P}^{(\tau)}$ defined by

$$\hat{P}_t^{(\tau)}((s',y') \mid (s,y),a) = \begin{cases} P(s' \mid s,a) \cdot (\mathbb{P}_{R(s,a)} \circ \phi^{-1})(y'-y) & \text{if } t \leq \tau \\ P(s' \mid s,a) \cdot \mathbb{P}_{R(s,a)}(y'-y) & \text{otherwise.} \end{cases}$$

Then, $\mathcal{M}_\tau$ is identical to $\mathcal{M}_{\tau-1}$ except $\mathbb{P}_{R(s,a)}$ is replaced with $\mathbb{P}_{R(s,a)} \circ \phi^{-1}$ on step $\tau$. We prove three lemmas showing a lower bound on the value of a policy $\pi$ for $\mathcal{M}_\tau$ when adapted to $\mathcal{M}_{\tau-1}$.

**Lemma B.1.** *Given $\tau \in [T]$, let $\mathcal{M} = \mathcal{M}_{\tau-1}$ and $\hat{\mathcal{M}} = \mathcal{M}_\tau$ (so compared to $\mathcal{M}$, $\hat{\mathcal{M}}$ replaces $\mathbb{P}_{R(s,a)}$ with $\mathbb{P}_{R(s,a)} \circ \phi^{-1}$ on step $\tau$ in its transitions). Given any policy $\hat{\pi}$ for $\hat{\mathcal{M}}$, define the policy*

$$\pi_t(a \mid s,y,\alpha) = \hat{\pi}_t(a \mid s,y+\alpha)$$

*for $\mathcal{M}$, where we initialize the (extra) policy internal state $\alpha_1 = 0$, and we update $\alpha_{\tau+1} = \phi(r_\tau) - r_\tau$ on step $\tau$ and $\alpha_{t+1} = \alpha_t$ otherwise. Then, for all $x,y,\alpha \in \mathbb{R}$, for $t > \tau$, we have*

$$F_{Z_t^{(\pi)}(s,y,\alpha)}(x) = F_{\hat{Z}_t^{(\hat{\pi})}(s,y+\alpha)}(x),$$

*and for $t \leq \tau$, we have*

$$F_{Z_t^{(\pi)}(s,y,0)}(x) \leq F_{\hat{Z}_t^{(\hat{\pi})}(s,y)}(x+\eta),$$

*where $Z_t^{(\pi)}$ (resp., $\hat{Z}_t^{(\hat{\pi})}$) is the return of $\mathcal{M}$ (resp., $\hat{\mathcal{M}}$) from step $t$ for policy $\pi$ (resp., $\hat{\pi}$).*

*Proof.* We prove by backwards induction on $t$. The base case $t = T$ follows by definition (and since the reward measure does not change from $\mathcal{M}$ to $\hat{\mathcal{M}}$). For $t > \tau$, we have

$$F_{Z_t^{(\pi)}(s,y,\alpha)}(x) = \sum_{a \in A} \sum_{s' \in S} \int \pi_t(a \mid s,y,\alpha) \cdot P(s' \mid s,a) \cdot F_{Z_{t+1}^{(\pi)}(s',y+r,\alpha)}(x-r) \cdot d\mathbb{P}_{R(s,a)}(r)$$

$$= \sum_{a \in A} \sum_{s' \in S} \int \hat{\pi}_t(a \mid s,y+\alpha) \cdot P(s' \mid s,a) \cdot F_{\hat{Z}_{t+1}^{(\hat{\pi})}(s',y+\alpha+r)}(x-r) \cdot d\mathbb{P}_{R(s,a)}(r)$$

$$= F_{\hat{Z}_t^{(\hat{\pi})}(s,y+\alpha)}(x),$$

where the second line follows by induction and by the definition of $\pi$. Next, for $t = \tau$, we have

$$F_{Z_t^{(\pi)}(s,y,0)}(x) = \sum_{a \in A} \sum_{s' \in S} \int \pi_t(a \mid s,y,0) \cdot P(s' \mid s,a) \cdot F_{Z_{t+1}^{(\pi)}(s',y+r,\phi(r)-r)}(x-r) \cdot d\mathbb{P}_{R(s,a)}(r)$$

$$= \sum_{a \in A} \sum_{s' \in S} \int \hat{\pi}_t(a \mid s,y) \cdot P(s' \mid s,a) \cdot F_{\hat{Z}_{t+1}^{(\hat{\pi})}(s',y+\phi(r))}(x-r) \cdot d\mathbb{P}_{R(s,a)}(r)$$

$$= \sum_{a \in A} \sum_{s' \in S} \int \hat{\pi}_t(a \mid s,y) \cdot P(s' \mid s,a) \cdot F_{\hat{Z}_{t+1}^{(\hat{\pi})}(s',y+\phi(r))}(x-\phi(r)+\phi(r)-r) \cdot d\mathbb{P}_{R(s,a)}(r)$$

$$\leq \sum_{a \in A} \sum_{s' \in S} \int \hat{\pi}_t(a \mid s,y) \cdot P(s' \mid s,a) \cdot F_{\hat{Z}_{t+1}^{(\hat{\pi})}(s',y+\phi(r))}(x-\phi(r)+\eta) \cdot d\mathbb{P}_{R(s,a)}(r)$$

$$= \sum_{a \in A} \sum_{s' \in S} \int \hat{\pi}_t(a \mid s,y) \cdot P(s' \mid s,a) \cdot F_{\hat{Z}_{t+1}^{(\hat{\pi})}(s',y+\rho)}(x-\rho+\eta) \cdot d\mathbb{P}_{R(s,a)} \circ \phi^{-1}(\rho)$$

$$= F_{\hat{Z}_t^{(\hat{\pi})}(s,y)}(x+\eta),$$

where the first line uses the update from $\alpha_t = 0$ to $\alpha_{t+1} = r - \phi(r)$ on this step, the second line follows by induction and by the definition of $\pi$, the fourth line follows by monotonicity of $F_{\hat{Z}_{t+1}^{(\hat{\pi})}(s',y+r)}$, and the fifth line follows by a change of variables $\rho = \phi(r)$. For $t < \tau$, we have

$$F_{Z_t^{(\pi)}(s,y,0)}(x) = \sum_{a \in A} \sum_{s' \in S} \int \pi_t(a \mid s,y,0) \cdot P(s' \mid s,a) \cdot F_{Z_{t+1}^{(\pi)}(s',y+r,0)}(x-r) \cdot d\mathbb{P}_{R(s,a)}(r)$$

$$= \sum_{a \in A} \sum_{s' \in S} \int \hat{\pi}_t(a \mid s,y) \cdot P(s' \mid s,a) \cdot F_{Z_{t+1}^{(\pi)}(s',y+r,0)}(x-r) \cdot d\mathbb{P}_{R(s,a)}(r)$$

$$\leq \sum_{a \in A} \sum_{s' \in S} \int \hat{\pi}_t(a \mid s,y) \cdot P(s' \mid s,a) \cdot F_{\hat{Z}_{t+1}^{(\hat{\pi})}(s',y+r,0)}(x-r+\eta) \cdot d\mathbb{P}_{R(s,a)}(r)$$

$$= F_{\hat{Z}_t^{(\hat{\pi})}(s',0)}(x+\eta),$$

where the second line follows by the definition of $\pi$, and the third line follows by induction. The claim follows. $\qquad\square$

**Lemma B.2.** *For any monotonically increasing $F$, we have $F^\dagger(F(x)) \leq x$, and $F(F^\dagger(\tau)) \geq \tau$.*

*Proof.* See Proposition 1 in [18]. $\qquad\square$

**Lemma B.3.** *Let $F, G : \mathbb{R} \to \mathbb{R}$ be monotonically increasing. If $F(x) \leq G(x+\eta)$ for all $x \in \mathbb{R}$, then we have $F^\dagger(\tau) \geq G^\dagger(\tau) - \eta$ for all $\tau \in \mathbb{R}$.*

*Proof.* By assumption, $G(x) \geq F(x-\eta)$. Substituting $x = F^\dagger(\tau) + \eta$ into this formula, we obtain

$$G(F^\dagger(\tau) + \eta) \geq F(F^\dagger(\tau)) \geq \tau,$$

where the second inequality follows by Lemma B.2. Also by Lemma B.2, since $G$ is monotonically increasing, so is $G^\dagger$, so we can apply $G^\dagger$ to each side of the inequality to obtain

$$G^\dagger(\tau) \leq G^\dagger(G(F^\dagger(\tau) + \eta)) \leq F^\dagger(\tau) + \eta,$$

where the second inequality follows by Lemma B.2. The claim follows. $\qquad\square$

**Lemma B.4.** *Consider the same setup as in Lemma B.1. Let $\hat{\pi}$ be a policy for $\hat{\mathcal{M}}$, and let $\pi$ be the policy defined in Lemma B.1 that adapts $\hat{\pi}$ to $\mathcal{M}$. Then, we have*

$$\Phi(\pi) \geq \hat{\Phi}(\hat{\pi}) - \eta,$$

*where $\Phi$ is the objective for $\mathcal{M}$ and $\hat{\Phi}$ is the objective for $\hat{\mathcal{M}}$.*

*Proof.* Let $\hat{Z} = \hat{Z}_1^{(\hat{\pi})}(s_1, 0)$ and $Z = Z_1^{(\pi)}(s_1, 0, 0)$. Applying Lemma B.3 to the inequality in Lemma B.1, we have

$$F_Z^\dagger(\tau) \geq F_{\hat{Z}}^\dagger(\tau) - \eta.$$

Integrating this inequality, we have

$$\Phi(\pi) = \int F_Z^\dagger(\tau) \cdot dG(\tau) \geq \int \left( F_{\hat{Z}}^\dagger(\tau) - \eta \right) \cdot dG(\tau) = \hat{\Phi}(\hat{\pi}) - \eta,$$

as claimed. $\qquad\square$

**Lemma B.5.** *Consider the same setup as in Lemma B.1. Given any policy $\pi$ for $\mathcal{M}$, define the policy*

$$\hat{\pi}_t(a \mid s,y,\alpha) = \pi_t(a \mid s,y+\alpha)$$

*for $\hat{\mathcal{M}}$, where we initialize $\alpha_1 = 0$, and we update $\alpha_{\tau+1} = r$ on step $\tau$, where $r$ is a random variable with probability measure*

$$\mathbb{P}_{R(s,a)}(r \mid \phi(r) = \rho),$$

*and $\alpha_{t+1} = \alpha_t$ otherwise. Then, for all $x,y,\alpha \in \mathbb{R}$, for $t > \tau$, we have*

$$F_{\hat{Z}_t^{(\hat{\pi})}(s,y,\alpha)}(x) = F_{Z_t^{(\pi)}(s,y+\alpha)}(x),$$

*and for $t \leq \tau$, we have*

$$F_{\hat{Z}_t^{(\hat{\pi})}(s,y,0)}(x) \leq F_{Z_t^{(\pi)}(s,y)}(x).$$

*Proof.* We prove by backwards induction on $T$. The base case $t = T$ follows by definition. For $t > \tau$, we have

$$F_{\hat{Z}_t^{(\hat{\pi})}(s,y,\alpha)}(x) = \sum_{a \in A} \sum_{s' \in S} \int \hat{\pi}_t(a \mid s, y, \alpha) \cdot P(s' \mid s, a) \cdot F_{\hat{Z}_{t+1}^{(\pi)}(s',y+\rho,\alpha)}(x - \rho) \cdot d\mathbb{P}_{R(s,a)}(\rho)$$

$$= \sum_{a \in A} \sum_{s' \in S} \int \pi_t(a \mid s, y + \alpha) \cdot P(s' \mid s, a) \cdot F_{Z_{t+1}^{(\pi)}(s',y+\alpha+\rho)}(x - \rho) \cdot d\mathbb{P}_{R(s,a)}(\rho)$$

$$= F_{Z_t^{(\pi)}(s,y+\alpha)}(x),$$

where the second line follows by induction and by the definition of $\pi$. Next, for $t = \tau$, we have

$$F_{\hat{Z}_t^{(\hat{\pi})}(s,c,0)}(x)$$

$$= \sum_{a \in A} \sum_{s' \in S} \int \hat{\pi}_t(a \mid s, y, 0) \cdot P(s' \mid s, a) \cdot F_{\hat{Z}_{t+1}^{(\hat{\pi})}(s',y+\rho,r-\rho)}(x - \rho) \cdot d\mathbb{P}_{R(s,a)}(r \mid \phi(r) = \rho) \cdot d\mathbb{P}_{R(s,a)} \circ \phi^{-1}(\rho)$$

$$= \sum_{a \in A} \sum_{s' \in S} \int \pi_t(a \mid s, y) \cdot P(s' \mid s, a) \cdot F_{Z_{t+1}^{(\pi)}(s',y+r)}(x - \rho) \cdot d\mathbb{P}_{R(s,a)}(r \mid \phi(r) = \rho) \cdot d\mathbb{P}_{R(s,a)} \circ \phi^{-1}(\rho)$$

$$\leq \sum_{a \in A} \sum_{s' \in S} \int \pi_t(a \mid s, y) \cdot P(s' \mid s, a) \cdot F_{Z_{t+1}^{(\pi)}(s',y+r)}(x - r) \cdot d\mathbb{P}_{R(s,a)}(r \mid \phi(r) = \rho) \cdot d\mathbb{P}_{R(s,a)} \circ \phi^{-1}(\rho)$$

$$= \sum_{a \in A} \sum_{s' \in S} \int \pi_t(a \mid s, y) \cdot P(s' \mid s, a) \cdot F_{Z_{t+1}^{(\pi)}(s',y+r)}(x - r) \cdot d\mathbb{P}_{R(s,a)}(r)$$

$$= F_{Z_t^{(\pi)}(s,y)}(x),$$

where the second line uses the update from $\alpha_t = 0$ to $\alpha_{t+1} = r - \rho$ on this step, the third line follows by induction and by the definition of $\hat{\pi}$, the fourth line follows by monotonicity of $F_{Z_{t+1}^{(\pi)}(s',y+r)}$, and the fifth line follows by the definition of conditional probability—in particular,

$$\int F_{Z_{t+1}^{(\pi)}(s',y+r)}(x - r) \cdot d\mathbb{P}_{R(s,a)}(r \mid \phi(r) = \rho) \cdot d\mathbb{P}_{R(s,a)} \circ \phi^{-1}(\rho)$$

$$= \int \frac{\int F_{Z_{t+1}^{(\pi)}(s',y+r)}(x - r) \cdot \mathbb{1}(r \in \phi^{-1}(\rho)) \cdot d\mathbb{P}_{R(s,a)}(r)}{\int \mathbb{1}(r' \in \phi^{-1}(\rho)) \cdot d\mathbb{P}_{R(s,a)}(r')} \cdot d\mathbb{P}_{R(s,a)} \circ \phi^{-1}(\rho)$$

$$= \int F_{Z_{t+1}^{(\pi)}(s',y+r)}(x - r) \int \frac{\cdot \mathbb{1}(r \in \phi^{-1}(\rho))}{\int \mathbb{1}(r' \in \phi^{-1}(\rho)) \cdot d\mathbb{P}_{R(s,a)}(r')} \cdot d\mathbb{P}_{R(s,a)} \circ \phi^{-1}(\rho) \cdot d\mathbb{P}_{R(s,a)}(r)$$

$$= \int F_{Z_{t+1}^{(\pi)}(s',y+r)}(x - r) \sum_{i=1}^{\infty} \frac{\cdot \mathbb{1}(r \in B_i)}{\mathbb{P}(R(s,a) \in B_i)} \cdot \mathbb{P}(R(s,a) \in B_i) \cdot d\mathbb{P}_{R(s,a)}(r)$$

$$= \int F_{Z_{t+1}^{(\pi)}(s',y+r)}(x - r) \cdot d\mathbb{R}_{R(s,a)}(r),$$

where in the third line, $B_i = (\eta \cdot (i - 1), \eta \cdot i]$. For $t < \tau$, we have

$$F_{\hat{Z}_t^{(\hat{\pi})}(s,y,0)}(x) = \sum_{a \in A} \sum_{s' \in S} \int \hat{\pi}_t(a \mid s, y, 0) \cdot P(s' \mid s, a) \cdot F_{\hat{Z}_{t+1}^{(\hat{\pi})}(s',y+\rho,0)}(x - \rho) \cdot d\mathbb{P}_{R(s,a)}(\rho)$$

$$= \sum_{a \in A} \sum_{s' \in S} \int \pi_t(a \mid s, y) \cdot P(s' \mid s, a) \cdot F_{\hat{Z}_{t+1}^{(\hat{\pi})}(s',y+\rho,0)}(x - \rho) \cdot d\mathbb{P}_{R(s,a)}(\rho)$$

$$\leq \sum_{a \in A} \sum_{s' \in S} \int \pi_t(a \mid s, y) \cdot P(s' \mid s, a) \cdot F_{Z_{t+1}^{(\pi)}(s',y+\rho,0)}(x - \rho) \cdot d\mathbb{P}_{R(s,a)}(\rho)$$

$$= F_{Z_t^{(\pi)}(s',y,0)}(x),$$

where the second line follows by the definition of $\pi$, and the third line follows by induction. The claim follows. $\qquad \square$

Next, we prove two lemmas showing a converse—namely, a lower bound on the value of a policy $\pi$ for $\mathcal{M}_{\tau-1}$ when adapted to $\mathcal{M}_\tau$.

**Lemma B.6.** *Consider the same setup as in Lemma B.1. Letting $\pi$ be a policy for $\mathcal{M}$, and $\hat{\pi}$ be the policy defined in Lemma B.5 that adapts $\pi$ to $\hat{\mathcal{M}}$. Then, we have*

$$\hat{\Phi}(\hat{\pi}) \geq \Phi(\pi),$$

*where $\hat{\Phi}$ is the objective for $\hat{\mathcal{M}}$ and $\Phi$ is the objective for $\mathcal{M}$.*

*Proof.* Let $Z = Z_1^{(\pi)}(s_1, 0)$ and $\hat{Z} = Z_1^{(\hat{\pi})}(s_1, 0, 0)$. Applying Lemma B.3 to the inequality in Lemma B.5, we have

$$F_{\hat{Z}}^\dagger(\tau) \geq F_Z^\dagger(\tau).$$

Integrating this inequality, we have

$$\hat{\Phi}(\hat{\pi}) = \int F_{\hat{Z}}^\dagger(\tau) \cdot dG(\tau) \geq \int F_Z^\dagger(\tau) \cdot dG(\tau) = \Phi(\pi),$$

as claimed. $\qquad\qquad\qquad\qquad\qquad\qquad\qquad\qquad\qquad\qquad\qquad\qquad\qquad\qquad\quad$ $\square$

Finally, we prove Theorem 3.2. Let $\pi_T^T$ be the optimal policy for $\mathcal{M}_T$, and let $\pi_\tau^T$ be the policy defined in Lemma B.4 adapting $\pi_\tau^T$ from $\mathcal{M}_\tau$ to $\mathcal{M}_{\tau-1}$ for each $\tau \in [T]$.

$$\Phi_0(\pi_0^T) \geq \Phi_1(\pi_1^T) - \eta \geq \Phi_2(\pi_2^T) \geq ... \geq \Phi_T(\pi_T^T) - T \cdot \eta,$$

where each inequality follows by Lemma B.4. Similarly, let $\pi_0^0$ be the optimal policy for $\mathcal{M}_0$, and let $\pi_\tau^0$ be the policy defined in Lemma B.6 adapting $\pi_{\tau-1}^0$ from $\mathcal{M}_{\tau-1}$ to $\mathcal{M}_\tau$. Then, we have

$$\Phi_T(\pi_T^0) \geq \Phi_{T-1}(\pi_{T-1}^0) \geq ... \geq \Phi_0(\pi_0^0),$$

where each inequality follows by Lemma B.6. Furthermore, by optimality of $\pi_T^T$ for $\Phi_T$, we also have $\Phi_T(\pi_T^T) \geq \Phi_T(\pi_T^0)$; together, these three inequalities imply

$$\Phi_0(\pi_0^T) \geq \Phi_0(\pi_0^0) - T \cdot \eta.$$

Finally, note that $\pi_0^0 = \pi_{\tilde{\mathcal{M}}}^*$ is the optimal policy for $\tilde{\mathcal{M}} = \mathcal{M}_0$, and $\pi_T^0 = \pi_{\hat{\mathcal{M}}}$ is $\pi_0^0$ adapted to $\hat{\mathcal{M}}$; also, $\Phi_0 = \Phi_{\tilde{\mathcal{M}}}$ is the objective for $\tilde{\mathcal{M}}$. Thus, we have

$$\Phi_{\tilde{\mathcal{M}}}(\pi_{\hat{\mathcal{M}}}) \geq \Phi_{\tilde{\mathcal{M}}}(\pi_{\tilde{\mathcal{M}}}^*) - T \cdot \eta.$$

By Theorem 3.1, the optimal policy for $\tilde{\mathcal{M}}$ equals the optimal history-dependent policy for the original MDP $\mathcal{M}$, so the claim follows. $\quad\square$

## C  Proof of Lemmas for Section 5

### C.1  Proof of Lemma 5.2

*Proof.* First, by the Dvoretzky–Kiefer–Wolfowitz (DKW) inequality and a union bound, for each $k \in [K]$, conditioned on $\{N^{(k)}(s, a)\}_{s \in \mathcal{S}, a \in \mathcal{A}}$, with probability at least $1 - \delta/(3K)$, we have

$$\|F_{\tilde{R}^{(k)}(s,a)} - F_{R(s,a)}\|_\infty \leq \epsilon_R^{(k)}(s, a) \qquad (\forall s \in \mathcal{S}, a \in \mathcal{A}).$$

Similarly, by Hoeffding's inequality, an $\ell_1$ concentration bound for multinomial distribution, and a union bound, for each $k \in [K]$, conditioned on $\{N^{(k)}(s, a)\}_{s \in \mathcal{S}, a \in \mathcal{A}}$, we have

$$\|\tilde{P}^{(k)}(\cdot \mid s, a) - P(\cdot \mid s, a)\|_1 \leq \epsilon_P^{(k)}(s, a) \quad (\forall s \in \mathcal{S}, a \in \mathcal{A})$$

$$\|\tilde{P}^{(k)}(\cdot \mid s, a) - P(\cdot \mid s, a)\|_\infty \leq \epsilon_R^{(k)}(s, a) \quad (\forall s \in \mathcal{S}, a \in \mathcal{A}).$$

each holding with probability at least $1 - \delta/(3K)$, respectively. Thus, both of these bounds hold for all $k \in [K]$ with probability at least $1 - \delta$. The claim follows. $\qquad\qquad\qquad\qquad$ $\square$

## C.2 Proof of Lemma 5.3

*Proof.* First, we prove that for all policies $\pi$, we have

$$\|F_{Z'^{(\pi)}} - F_{Z^{(\pi)}}\|_\infty \le B(\pi).$$

To this end, let $G(r) = F_{Z_{t+1}^{(\pi)}(s',y+r)}(x - r)$; note that $G(-\infty) = 1$ and $G(\infty) = 0$. Then, by integration by parts, we have

$$F_{Z_t^{(\pi)}(s,y)}(x) = \int \sum_{a \in \mathcal{A}} \sum_{s' \in \mathcal{S}} \pi(a \mid s,y) \cdot P(s' \mid s,a) \cdot G(r) \cdot d\mathbb{P}_{R(s,a)}(r)$$

$$= -\int \sum_{a \in \mathcal{A}} \sum_{s' \in \mathcal{S}} \pi(a \mid s,y) \cdot P(s' \mid s,a) \cdot F_{R(s,a)}(r) \cdot dG(r),$$

and similarly for $F_{Z_t'^{(\pi)}(s,y)}(x)$. Next, note that

$$\sup_{x \in \mathbb{R}} |F_{Z_t'^{(\pi)}(s,y)}(x) - F_{Z_t^{(\pi)}(s,y)}(x)|$$

$$= \sup_{x \in \mathbb{R}} \left| \sum_{a \in \mathcal{A}} \sum_{s' \in \mathcal{S}} \pi(a \mid s,y) \left(P'(s' \mid s,a) - P(s' \mid s,a)\right) \int F_{Z_{t+1}'^{(\pi)}(s',y+r)}(x - r) d\mathbb{P}'_{R'(s,a)}(r) \right.$$

$$+ \sum_{a \in \mathcal{A}} \sum_{s' \in \mathcal{S}} \pi(a \mid s,y) P(s' \mid s,a) \int \left( F_{Z_{t+1}'^{(\pi)}(s',y+r)}(x - r) - F_{Z_{t+1}^{(\pi)}(s',y+r)}(x - r) \right) d\mathbb{P}'_{R'(s,a)}(r)$$

$$\left. - \sum_{a \in \mathcal{A}} \sum_{s' \in \mathcal{S}} \pi(a \mid s,y) P(s' \mid s,a) \int \left( F_{R'(s,a)}(r) - F_{R(s,a)}(r) \right) dF_{Z_{t+1}^{(\pi)}(s',y+r)}(x - r) \right|$$

$$\le \sup_{x \in \mathbb{R}} \sum_{a \in \mathcal{A}} \sum_{s' \in \mathcal{S}} \pi(a \mid s,y) \cdot |P'(s' \mid s,a) - P(s' \mid s,a)|$$

$$+ \sum_{a \in \mathcal{A}} \sum_{s' \in \mathcal{S}} \pi(a \mid s,y) \cdot \int \sup_{x' \in \mathbb{R}} |F_{Z_{t+1}'^{(\pi)}(s',y+r)}(x') - F_{Z_{t+1}^{(\pi)}(s',y+r)}(x')| \cdot d\mathbb{P}'_{R'(s,a)}(r)$$

$$+ \sum_{a \in \mathcal{A}} \pi(a \mid s,y) \cdot \sup_{r' \in \mathbb{R}} |F_{R'(s,a)}(r') - F_{R(s,a)}(r')|$$

$$\le \mathbb{E}\left[ \epsilon_P(s,a) + \epsilon_R(s,a) + \sup_{x' \in \mathbb{R}} |F_{Z_{t+1}'^{(\pi)}(s',y+r)}(x') - F_{Z_{t+1}^{(\pi)}(s',y+r)}(x')| \right].$$

Thus, we have

$$\epsilon_t^{(\pi)} := \mathbb{E}\left[ \sup_{x \in \mathbb{R}} |F_{Z_t'^{(\pi)}(s,y)}(x) - F_{Z_t^{(\pi)}(s,y)}(x)| \right]$$

$$= \mathbb{E}\left[ \epsilon_{s,a}^{(k,P)} + \epsilon_{s,a}^{(k,R)} + \sup_{x' \in \mathbb{R}} |F_{Z_{t+1}'^{(\pi)}(s',y+r)}(x') - F_{Z_{t+1}^{(\pi)}(s',y+r)}(x')| \right]$$

$$\le \mathbb{E}\left[ \epsilon_P(s,a) + \epsilon_R(s,a) \right] + \epsilon_{t+1}^{(k,\pi)}$$

$$= \mathbb{E}\left[ \sum_{\tau=t}^{T} \epsilon_P(s_\tau, a_\tau) + \epsilon_R(s_\tau, a_\tau) \right],$$

where the last step follows by induction. Finally, we have

$$|\Phi'(\pi) - \Phi(\pi)| = \left| \int_0^T \left( G(F_{Z'^{(\pi)}}(x)) - G(F_{Z^{(\pi)}}(x)) \right) \cdot dx \right|$$

$$\le L_G \int_0^T |F_{Z'^{(\pi)}}(x) - F_{Z^{(\pi)}}(x)| \cdot dx$$

$$\le T \cdot L_G \cdot \epsilon_1^{(\pi)},$$

where the first line follows by Lemma 5.1. The claim follows since $\epsilon_1^{(\pi)}$ equals the desired bound. $\square$

## C.3 Proof of Lemma 5.7

*Proof.* First, we prove that $F_{\hat{Z}_t^{(k,\pi)}(s,y)}(x) \leq F_{Z_t^{(\pi)}(s,y)}(x)$. The case $s = s_\infty$ is straightforward, since its transitions and rewards are equal in $\mathcal{M}$ and $\hat{\mathcal{M}}$, and it only transitions to itself. For $s \neq s_\infty$, we prove by induction on $t$. The base case $t = T$ follows by definition. Then, we have

$$
\begin{aligned}
F_{\hat{Z}_t^{(k,\pi)}(s,y)}(x) &= \int \sum_{a \in \mathcal{A}} \sum_{s' \in \mathcal{S}} \pi(a \mid s, y) \cdot \hat{P}^{(k)}(s' \mid s, a) \cdot F_{\hat{Z}_{t+1}^{(k,\pi)}(s',y+r)}(x-r) \cdot d\hat{\mathbb{P}}_{\hat{R}^{(k)}(s,a)}(r) \\
&\leq \int \sum_{a \in \mathcal{A}} \sum_{s' \in \mathcal{S}} \pi(a \mid s, y) \cdot \hat{P}^{(k)}(s' \mid s, a) \cdot F_{Z_{t+1}^{(\pi)}(s',y+r)}(x-r) \cdot d\hat{\mathbb{P}}_{\hat{R}^{(k)}(s,a)}(r) \\
&= \int \sum_{a \in \mathcal{A}} \sum_{s' \in \mathcal{S}} \pi(a \mid s, y) \cdot \hat{P}^{(k)}(s' \mid s, a) \cdot F_{\hat{R}^{(k)}(s,a)}(x' - x) \cdot dF_{Z_{t+1}^{(\pi)}(s',y+r)}(x') \\
&\leq \int \sum_{a \in \mathcal{A}} \sum_{s' \in \mathcal{S}} \pi(a \mid s, y) \cdot \hat{P}^{(k)}(s' \mid s, a) \cdot F_{R(s,a)}(x' - x) \cdot dF_{Z_{t+1}^{(\pi)}(s',y+r)}(x') \\
&= \int \sum_{a \in \mathcal{A}} \sum_{s' \in \mathcal{S}} \pi(a \mid s, y) \cdot \hat{P}^{(k)}(s' \mid s, a) \cdot F_{Z_{t+1}^{(\pi)}(s',y+r)}(x-r) \cdot d\mathbb{P}_{R(s,a)}(r),
\end{aligned}
\tag{2}
$$

where the second line follows by induction, the third by integration by parts and substituting $x' = x - r$, the fourth since $F_{R(s,a)}(r) = 1 = F_{\hat{R}^{(k)}(s,a)}(r)$ for $r \geq 1$, and for $r < 1$, on event $\mathcal{E}$, we have

$$
F_{R(s,a)}(r) \geq \max\left\{ F_{\tilde{R}^{(k)}(s,a)}(r) - \epsilon_R^{(k)}(s,a), 0 \right\} = F_{\hat{R}^{(k)}(s,a)}(r),
$$

and the fifth by integration by parts and substituting $r = x' - x$. Next, since $s \neq s_\infty$, we have

$$
\hat{P}^{(k)}(s_\infty \mid s, a) = 1 - \sum_{s' \in \mathcal{S} \setminus \{s_\infty\}} \hat{P}^{(k)}(s' \mid s, a) = \sum_{s' \in \mathcal{S} \setminus \{s_\infty\}} P(s' \mid s, a) - \hat{P}^{(k)}(s' \mid s, a),
$$

so we can decompose the summand $\hat{P}^{(k)}(s_\infty \mid s, a) \cdot F_{Z_{t+1}^{(\pi)}(s_\infty,y+r)}(x-r)$ (i.e., $s' = s_\infty$) in (2) and distribute it across the other summands; in particular, the summands $s' \neq s_\infty$ become

$$
\begin{aligned}
&\hat{P}^{(k)}(s' \mid s, a) \cdot F_{Z_{t+1}^{(\pi)}(s',y+r)}(x-r) + \left( P(s' \mid s, a) - \hat{P}^{(k)}(s' \mid s, a) \right) \cdot F_{Z_{t+1}^{(\pi)}(s_\infty,y+r)}(x-r) \\
&\leq \hat{P}^{(k)}(s' \mid s, a) \cdot F_{Z_{t+1}^{(\pi)}(s',y+r)}(x-r) + \left( P(s' \mid s, a) - \hat{P}^{(k)}(s' \mid s, a) \right) \cdot F_{Z_{t+1}^{(\pi)}(s',y+r)}(x-r) \\
&= P(s' \mid s, a) \cdot F_{Z_{t+1}^{(\pi)}(s',y+r)}(x-r),
\end{aligned}
\tag{3}
$$

where the second line follows since $F_{Z_{t+1}^{(\pi)}(s_\infty,y+r)}(x-r) \leq F_{Z_{t+1}^{(\pi)}(s',y+r)}(x-r)$ for all $s' \neq s_\infty$, and since $P(s' \mid s, a) - \hat{P}^{(k)}(s' \mid s, a) \geq 0$ on event $\mathcal{E}$. Continuing from (2), we have

$$
\begin{aligned}
F_{\hat{Z}_t^{(k,\pi)}(s,y)}(x) &\leq \int \sum_{a \in \mathcal{A}} \sum_{s' \in \mathcal{S}} \pi(a \mid s, y) \cdot \hat{P}^{(k)}(s' \mid s, a) \cdot F_{Z_{t+1}^{(\pi)}(s',y+r)}(x-r) \cdot d\mathbb{P}_{R(s,a)}(r) \\
&\leq \int \sum_{a \in \mathcal{A}} \sum_{s' \in \mathcal{S} \setminus \{s_\infty\}} \pi(a \mid s, y) \cdot P(s' \mid s, a) \cdot F_{Z_{t+1}^{(\pi)}(s',y+r)}(x-r) \cdot d\mathbb{P}_{R(s,a)}(r) \\
&= F_{Z_t^{(\pi)}(s,y)}(x),
\end{aligned}
\tag{4}
$$

where the second line follows by distributing the summand $s' = s_\infty$ and applying (3). Since $\mathcal{M}$ and $\hat{\mathcal{M}}$ have the same initial state distribution, we have $F_{\hat{Z}^{(k,\pi)}}(x) \leq F_{Z^{(\pi)}}(x)$. By Lemma 5.1, we have

$$
\hat{\Phi}^{(k)}(\pi) = T - \int_0^T G(F_{\hat{Z}^{(k,\pi)}}(x)) \cdot dx \geq T - \int_0^T G(F_{Z^{(\pi)}}(x)) \cdot dx = \Phi(\pi),
$$

where the inequality follows from (4) and since $G$ is monotone. The claim follows. $\qquad\square$

## C.4 Proof of Theorem 4.1

*Proof.* We prove Theorem 4.1. Note that on event $\mathcal{E}$, we have

$\text{regret}(\mathfrak{A})$

$$= \sum_{k=1}^{K} \Phi(\pi^*) - \Phi(\hat{\pi}^{(k)})$$

$$\leq \sum_{k=1}^{K} \hat{\Phi}^{(k)}(\pi^*) - \Phi(\hat{\pi}^{(k)})$$

$$\leq \sum_{k=1}^{K} \hat{\Phi}^{(k)}(\hat{\pi}^{(k)}) - \Phi(\hat{\pi}^{(k)})$$

$$\leq \sum_{k=1}^{K} 2T \cdot L_G \cdot \sqrt{|\mathcal{S}|} \cdot B^{(k)}(\hat{\pi}^{(k)})$$

$$= 2TL_G \sqrt{5|\mathcal{S}|^2 \log\left(\frac{4|\mathcal{S}| \cdot |\mathcal{A}| \cdot K}{\delta}\right)} \cdot \mathbb{E}_{\Xi_T^{(\hat{\pi}^{(1:K)})}}\left[\sum_{k=1}^{K}\sum_{t=1}^{T} \frac{1}{\sqrt{N^{(k)}(s_t, a_t)}} \,\middle|\, \{N^{(k)}(s,a)\}_{k\in[K], s\in\mathcal{S}, a\in\mathcal{A}}\right],$$

where the first inequality follows by Lemma 5.7, the second follows by optimality of $\hat{\pi}^{(k)}$ for $\hat{\Phi}^{(k)}$, and the third by Lemma 5.6. Furthermore, note that

$$\sum_{k=1}^{K}\sum_{t=1}^{T} \frac{1}{\sqrt{N^{(k)}(s_t, a_t)}} \leq \sum_{k=1}^{K}\sum_{t=1}^{T} \mathbb{1}(N^{(k)}(s_t, a_t) \leq T) + \sum_{k=1}^{K}\sum_{t=1}^{T} \mathbb{1}(N^{(k)}(s_t, a_t) > T)\frac{1}{\sqrt{N^{(k)}(s_t, a_t)}}.$$

The event $(s_t, a_t) = (s, a)$ and $(N^{(k)}(s, a) \leq T)$ can happen fewer than $2T$ times per state action pair. Therefore, $\sum_{k=1}^{K}\sum_{t=1}^{T} \mathbb{1}(N^{(k)}(s_t, a_t) \leq T) \leq 2TSA$. Now suppose $N^{(k)}(s, a) > T$. Then for any $t \in \mathcal{W}_k$, we have $N_t^{(k)}(s, a) \leq N^{(k)}(s, a) + T \leq 2N^{(k)}(s, a)$. Thus, we have

$$\sum_{k=1}^{K}\sum_{t=1}^{T} \frac{\mathbb{1}(N^{(k)}(s_t, a_t) > T)}{\sqrt{N^{(k)}(s_t, a_t)}} \leq \sum_{k=1}^{K}\sum_{t=1}^{T} \sqrt{\frac{2}{N_t^{(k)}(s_t, a_t)}}$$

$$= \sqrt{2}\sum_{k=1}^{K}\sum_{t=1}^{T}\sum_{s\in\mathcal{S}}\sum_{a\in\mathcal{A}} \frac{\mathbb{1}((s_t, a_t) = (s, a))}{\sqrt{N_t^{(k)}(s, a)}}$$

$$\leq \sqrt{2}\sum_{s\in\mathcal{S}}\sum_{a\in\mathcal{A}} \sum_{j=1}^{N^{(K+1)}(s,a)} j^{-1/2}$$

$$\leq \sqrt{2}\sum_{s\in\mathcal{S}}\sum_{a\in\mathcal{A}} \int_{x=0}^{N^{(K+1)}(s,a)} x^{-1/2}dx$$

$$\leq \sqrt{2|\mathcal{S}| \cdot |\mathcal{A}| \cdot \sum_{s\in\mathcal{S}}\sum_{a\in\mathcal{A}} N^{(K+1)}(s,a)}$$

$$= \sqrt{2|\mathcal{S}| \cdot |\mathcal{A}| \cdot KT}.$$

The claim follows. □

# D  The Optimal Policy for CVaR Objectives

In this section, we describe how to compute the optimal policy for the CVaR objective when the MDP is known; this approach is described in detail in [13]. Following this work, we consider the setting where we are trying to minimize cost rather than maximize reward. In particular, consider an MDP $\mathcal{M} = (\mathcal{S}, \mathcal{A}, D, P, \mathbb{P}, T)$, and our goal is to compute a policy $\pi$ that maximizes its CVaR objective.

**Step 1: CVaR objective.** We begin by rewriting the CVaR objective in a form that is more amenable to optimization. First, we have the following key result (see [13] for a proof):

**Lemma D.1.** *For any random variable $Z$, we have*

$$CVaR_\alpha(Z) = \inf_{\rho \in \mathbb{R}} \left\{ \rho + \frac{1}{1-\alpha} \cdot \mathbb{E}_Z \left[ (Z - \rho)^+ \right] \right\},$$

*where the minimum is achieved by $\rho^* = VaR(Z)$.*

As a consequence of this lemma, we have

$$\min_{\pi \in \Pi} CVaR(Z^{(\pi)}) = \min_{\pi \in \Pi} \inf_{\rho \in \mathbb{R}} \left\{ \rho + \frac{1}{1-\alpha} \cdot \mathbb{E}_{Z^{(\pi)}} \left[ (Z^{(\pi)} - \rho)^+ \right] \right\}$$

$$= \inf_{\rho \in \mathbb{R}} \left\{ \rho + \frac{1}{1-\alpha} \cdot \min_{\pi \in \Pi} \mathbb{E}_{Z^{(\pi)}} \left[ (Z^{(\pi)} - \rho)^+ \right] \right\}.$$

Thus, we have

$$\pi^* = \arg\min_{\pi \in \Pi} \mathbb{E}_{Z^{(\pi)}} \left[ (Z^{(\pi)} - \rho^*)^+ \right],$$

where

$$\rho^* = \arg\inf_{\rho \in \mathbb{R}} J(\rho) \qquad \text{where} \qquad J(\rho) = \rho + \frac{1}{1-\alpha} \cdot \max_{\pi \in \Pi} \mathbb{E}_{Z^{(\pi)}} \left[ (Z^{(\pi)} - \rho)^+ \right].$$

The main challenge is evaluating the minimum over $\pi \in \Pi$ in $J(\rho)$. To do so, we construct another MDP whose objective is $\mathbb{E}_{Z^{(\pi)}} \left[ (Z^{(\pi)} - \rho)^+ \right]$ for the appropriate choice of initial state distribution.

**Step 2: Construct alternative MDP.** The MDP we construct is $\tilde{\mathcal{M}} = (\tilde{\mathcal{S}}, \mathcal{A}, \tilde{D}, \tilde{P}, \tilde{R}, T)$, where the states are $\tilde{\mathcal{S}} = \mathcal{S} \times \mathbb{R}$, the (time-varying, deterministic) rewards $\tilde{R} : \tilde{\mathcal{S}} \times [T] \to \mathbb{R}$ are

$$\tilde{R}((s, r), t) = \begin{cases} \max\{r, 0\} & \text{if } t = T \\ 0 & \text{otherwise,} \end{cases}$$

and the transitions are

$$\tilde{P}((s', r') \mid (s, r), a) = P(s' \mid s, a) \times \mathbb{P}_{R(s,a)}(r' - r),$$

noting that $\tilde{P}$ is a (conditional) probability measure since the state space $\tilde{\mathcal{S}}$ includes a continuous component; in practice, we discretize the continuous component of the state space.

**Step 3: Value iteration.** Letting $S_1$ be the random initial state of the original MDP $\mathcal{M}$ (with distribution $D$), we have

$$\mathbb{E}_{Z^{(\pi)}} \left[ (Z^{(\pi)} - \rho)^+ \right] = \mathbb{E}_{S_1} \left[ \tilde{V}_1^{(\pi)}((S_1, -\rho)) \right],$$

where $\tilde{V}_1^{(\pi)}$ is the value function of policy $\pi$ for MDP $\tilde{\mathcal{M}}$ on step $t = 1$. Thus, we have

$$\min_{\pi \in \Pi} \mathbb{E}_{Z^{(\pi)}} \left[ (Z^{(\pi)} - \rho)^+ \right] = \mathbb{E}_{S_1} \left[ \tilde{V}_1^*((S_1, -\rho)) \right],$$

where $\tilde{V}_1^*$ is the value function of the optimal policy for $\tilde{\mathcal{M}}$. Intuitively, this strategy works because the augmented component of the state space $r$ captures the cumulative reward so far plus its initial value $-\rho$; then, by the definition of $\tilde{R}$, the reward is $r^+$, which implies that $\tilde{V}_1^{(\pi)}((s, -\rho))$ is the expectation of the random variable $(Z^{(\pi)} - \rho)^+$. Thus, we can compute $\min_{\pi \in \Pi} \mathbb{E}_{Z^{(\pi)}}[(Z^{(\pi)} - \rho)^+]$ by performing value iteration on $\tilde{\mathcal{M}}$ to compute $\tilde{V}_1^{(\pi)}$. In particular, we have

$$\tilde{V}_T^*((s, r)) = \max\{r, 0\},$$

and

$$\tilde{V}_t^*((s, r)) = \min_{a \in A} \int \tilde{V}_{t+1}^*((s', r')) \cdot d\tilde{P}((s', r') \mid (s, r), a)$$

for all $t \in \{1, ..., T-1\}$. Then, given an initial state $s_1$, we construct state $\tilde{s}_1 = (s_1, -\rho^*)$, where

$$\rho^* = \arg\inf_{\rho \in \mathbb{R}} \left\{ \rho + \frac{1}{1-\alpha} \cdot \tilde{V}_1^{(\pi)}((s, -\rho)) \right\},$$

and then acting optimally in $\tilde{\mathcal{M}}$ according to $\tilde{V}_t^*$.