# OpenReview forum: "Regret Bounds for Risk-Sensitive Reinforcement Learning"
_NeurIPS.cc/2022/Conference — NeurIPS 2022 Accept_

### Official Review · Reviewer_wVQ7 · 2022-07-11

**Rating:** 7
**Confidence:** 2
**Soundness:** 3 good
**Presentation:** 4 excellent
**Contribution:** 3 good

**Summary:**


This paper describes an optimistic algorithm for a general class of Risk-Sensitive RL, where the quantiles of the return distribution are weighted according to a function G.
The paper proves a regret-bound linear with respect to Lipschitz constant $L_G$ of the weight function.

**Questions:**


- Between lines 113 and 114 should the left hand side of the equation be $\max_{\pi \in \Pi_{\text{Ind}}}\Phi_{\mathcal{\tilde{M}}}(\pi)$?
- In the right-hand side of the bottom equation after line 193, should it be $\epsilon_{R}^{(k)}$?

**Limitations:**


The paper clearly states the assumptions necessary to achieve the regret bounds claimed.

**Strengths And Weaknesses:**


### Strenghts

- The paper provides a novel analysis of a general class of Risk-sensitive RL problems.
- The paper is well written and well structured. The main results are proved in the main document and only the support material is left in the appendix.
- Overall, the paper provides an excellent presentation of the problem and the derivations.

### Weaknesses

- The motivation to study this general class of problems could be expanded to clarify further the significance of the proposed approach.

---

> ### Author Response · Authors · 2022-08-02
> **Response to Reviewer wVQ7**
>
> Thank you for your comments and feedback.
>
> **Motivation.** First, we note that providing a regret bound for the popular CVaR objective is already an important contribution. We use the more general class of problems since our proof strategy extends naturally to this more general class. We note that other risk sensitive-objectives can also be captured in this framework, for example the Wang measure [1], and the cumulative probability weighting (CPW) metric [2].
>
> **Lines 113-114.** Actually the current form of the equation is correct, as we are arguing that only considering history-independent policies in the augmented MDP $\tilde{\mathcal{M}}$ gives the same optimal objective value as optimizing over any policy in the original MDP $\mathcal{M}$. In fact, these two optimal objective values coincide due to Theorem 3.1.
>
> **Line 193.** Indeed, thanks for pointing this out. We will fix the typo.
>
> **References.**
>
> [1] Shaun S Wang. A Class of Distortion Operators for Pricing Financial and Insurance Risks. *Journal of Risk and Insurance*, pages 15–36, 2000.
>
> [2] Amos Tversky and Daniel Kahneman. Advances in Prospect Theory: Cumulative Representation of Uncertainty. *Journal of Risk and Uncertainty*, 5(4):297–323, 1992.

---

> > ### Comment · Reviewer_wVQ7 · 2022-08-07
> > **Thanks to the authors and clarification regarding lines 113-114**
> >
> > I would like to thank the authors for addressing my concern regarding the motivation of the paper.
> >
> > Perhaps I could clarify my point regarding Lines 113-114 now(128-129). This equation seems to be missing the argument of the function $\Phi$ which according to Eq 1 gets a policy as input on the form: $\Phi(\pi)$.
> >
> > Please let me know if I misunderstood the usage of $\Phi$ on this equation.

---

> > > ### Author Response · Authors · 2022-08-07
> > > **Thanks to the reviewer and response regarding lines 113-114**
> > >
> > > Thank you for your clarification. You are correct, this is a typo, and the argument $\pi$ should be present; thank you for pointing it out! We have updated our draft (currently between lines 128-129) to fix this.

---

### Official Review · Reviewer_TTmD · 2022-07-11

**Rating:** 7
**Confidence:** 3
**Soundness:** 3 good
**Presentation:** 2 fair
**Contribution:** 4 excellent

**Summary:**

This paper presented a novel regret bound analysis for risk-sensitive reinforcement learning. The analysis is based on an optimistic MDP formulation. The regret bound could be applied to more general problem formulations compared to existing regret bound analyses. Overall, I am not quite familiar with the related work of regret bounds of the risk-sensitive RL, and the authors also did not explain too much about related works in the main paper. Therefore, I am not sure if the claim of the first regret bound is right. However, the results are reasonable on CVaR problem formulation and seem plausible. Therefore, I have a positive impression of this paper. I also suggest that the author could explain more about related work (even though there might not be many) to help better understand this paper.

**Questions:**

1. It seems that the results rely heavily on the Lipschitz constant $L_G$. What if the $L_G$ is not Lipschitz?
2. Are there any other requirements of the MDP, such as stationary or non-stationary?

**Limitations:**

The author adequately addressed the limitations and there is no potential social impact of this work.

**Strengths And Weaknesses:**

Strength:
The paper gave a thorough analysis of regret bound of risk-sensitive reinforcement learning. The notions are clear and the authors also explained many intuitions behind the proof.

Weakness:
The claim of the first regret bound might be too strong, and the logic of the whole paper could be further improved (e.g., there are many lemmas placed at the end that makes me confused initially. )

---

> ### Author Response · Authors · 2022-08-02
> **Response to Reviewer TTmD**
>
> Thank you for your comments and feedback.
>
> **Lipschitz assumption.** We believe that the function $G$ being Lipschitz is crucial to achieving the desired regret guarantee. Intuitively, if $G$ is discontinuous, then the distribution $g(\tau)$ is a delta function at the discontinuity. However, any nonzero error in our estimate of the underlying MDP can perturb the return at $\tau$. As a consequence, we cannot guarantee computing the optimal policy unless we know $M$ exactly, meaning any algorithm may achieve linear regret.
>
> **MDP.** All the requirements of the MDP are stated in Section 2 *Problem Formulation*; in particular, we implicitly define the MDP to be stationary. However, we note that our analysis easily extends to MDPs $M_1,...,M_T$ that vary across time steps within an episode, as long as these MDPs are stationary across episodes.
>
> **Novelty.** We have clarified the novelty of our work by adding a related work section; at a high level, we are not aware of any prior work that provides regret bounds for RL with CVaR objectives.

---

> > ### Comment · Reviewer_TTmD · 2022-08-09
> > **Response to Paper7137 Authors**
> >
> > Thank the authors for addressing my concerns! All the questions are appropriately explained. I do not have any further questions.

---

### Official Review · Reviewer_bBTj · 2022-07-11

**Rating:** 7
**Confidence:** 4
**Soundness:** 3 good
**Presentation:** 3 good
**Contribution:** 3 good

**Summary:**

This paper proposes an exploration learning algorithm for risk-sensitive reinforcement learning algorithm with provable guarantees on its performance using the metric of regret.

**Questions:**

Please see "Weaknesses' above.

**Limitations:**

Limitations are specified in the form of the assumptions.

**Strengths And Weaknesses:**

Strengths:

1. This paper presents a general approach for addressing the risk-sensitive reinforcement learning problems and use this approach to get provable regret guarantees. In particular, the paper considers the general formulation given Eq. (1). One of the most popular risk-sensitive criteria CVaR will become a special case of this formulation.

2. According to this reviewer, the key contribution of this paper is the result given in Lemma 5.1. Risk-sensitive RL is difficult mainly because of the complicated form of the objective function. Lemma 5.1 converts this to a clearly tractably form, which can be used to bound the difference in the objective function due to two policies by the difference in quantile functions of the cumulative rewards induced by those policies. This result is possibly useful in many other research problems, including in the analysis of policy gradient algorithms.

3. The paper also presents a new style of UCB algorithm which is different from the standard optimism in the face of uncertainty (OFU) algorithm. Most of the standard algorithms finds an optimistic model by searching within a ball around the empirical estimate of the model in such a way to maximize the objective. This paper seems to be directly selecting a model by subtracting a penalty term (corresponding to the visitation counts) from the estimated model. This will reduce the computational complexity compared to the standard OFU algorithm which requires approaches such as extended value iteration.



Weaknesses:

1. Proofs are rather compressed. Please provide intuitive explanation general proof ideas. Also, please provide explanation for the intermediate steps.
2. The motivation for using a different style of UCB approach is not clear. Is this approach used in prior works? What are the advantages or disadvantages of this approach?
3. The computational feasibility of the proposed generalized form is not clear. Even for the CVaR setting, how is the oracle policy computed? For general setting, is there a standard algorithm?
4. This reviewer acknowledges that this is a theoretical paper. However, given that it proposes a new algorithm, and deals only with tabular setting, it is important to include the experimental results which demonstrate the performance of the algorithm.

---

> ### Author Response · Authors · 2022-08-02
> **Response to Reviewer bBTj**
>
> Thank you for your comments and feedback.
>
> **Proof sketch.** We will add an expanded version of the following proof sketch at the beginning of Section 5. At a high level, the proof proceeds in three steps. First, we prove our key Lemma 5.1 that expresses the objective function $\Phi$ in terms of an integral of the weighted CDF of the return. This lemma allows us to translate bounds on the difference between CDFs of the estimated return $\hat{Z}^{(\pi)}$ and the true return $Z^{(\pi)}$ into bounds on the difference between corresponding objective values. The proof of this lemma is divided into three parts that deal with different sets of points in the domain of the quantile function $F_{Z^{(\pi)}}^\dagger$: (i) discontinuous; (ii) continuous and strictly monotone; (iii) continuous and non-strictly monotone.
>
> Second, we define a high probability event $\mathcal{E}$ in Lemma 5.2 where the empirically estimated MDP $\hat{\mathcal{M}}^{(k)}$ falls into a confidence set around the true discretized augmented MDP $\mathcal{M}^{(k)}$. Under $\mathcal{E}$, we prove that the objective values of $\hat{\mathcal{M}}^{(k)}$ and $\mathcal{M}^{(k)}$ are close in Lemma 5.6 by showing separately that (i) the objective values of the empirical MDP $\tilde{\mathcal{M}}^{(k)}$ and $\mathcal{M}^{(k)}$ are close in Lemma 5.4, and (ii) the objective values of $\hat{\mathcal{M}}^{(k)}$ and $\tilde{\mathcal{M}}^{(k)}$ are close in Lemma 5.5.
>
> Finally, we prove that the optimistic MDP $\hat{\mathcal{M}}^{(k)}$ is indeed optimistic under the high probability event $\mathcal{E}$ in Lemma 5.7. Together, these results imply the regret bound using the standard UCB proof strategy.
>
> **Algorithmic contribution.** The UCBVI algorithm [1] constructs an optimistic MDP by adding a bonus to the reward of a state-action pair that is inversely proportional to its uncertainty. However, this is not suitable for maximizing the risk-sensitive objective, which requires optimism over the return distribution. The extra state $s_\infty$ in our construction is crucial to dealing with this issue.
>
> In addition, the extended value iteration algorithm used in UCRL2 [2] also does not work here because we assume a blackbox oracle for a given MDP. That is, we cannot search in a set of possible MDPs for the most optimistic one, but instead need to construct an explicit optimistic MDP for the oracle to compute an optimal policy.
>
> **Oracle policy.** For CVaR objectives, the oracle policy can be computed via standard value iteration on an augmented MDP [3]. We will include an explanation of the procedure in the appendix. To the best of our knowledge, there is no known standard algorithm to compute the optimal policy for general quantile-based risk-sensitive objectives. However, we emphasize that we provide a complete solution in the important case of CVaR --- i.e., an efficient learning algorithm together with a regret analysis.
>
> **Experiments.** While we are happy to run simulations of our algorithm on a tabular environment, we are not sure what would be a suitable baseline. For instance, the optimism strategy and extended value iteration used in UCRL2 do not work in our setting, nor does the optimism strategy used in UCBVI.
>
> **References.**
>
> [1] Azar et al., Minimax Regret Bounds for Reinforcement Learning. *International Conference on Machine Learning*, pages 263-272. PMLR, 2017.
>
> [2] Auer et al., Near-Optimal Regret Bounds for Reinforcement Learning. *Advances in Neural Information Processing Systems*, 21, 2008.
>
> [3] Bäuerle and Ott. Markov Decision Processes with Average-Value-at-Risk Criteria. *Mathematical Methods of Operations Research*, 74(3):361-379, 2011.

---

> > ### Comment · Reviewer_bBTj · 2022-08-08
> > **Thanks to the authors**
> >
> > Thank you for giving satisfactory response to my comments.
> >
> > 1.About the simulation: You should have an empirical experiments section in your paper, given that you are proposing a new algorithm. Lack of a suitable baseline is not a satisfactory argument for not including an experimental demonstration of a new proposed algorithm. The primary goal of the experiments section is to give evidence that the proposed approach works as expected. Now, regarding the baseline, you can compare the performance against the oracle solution. Also, you can compare the performance against a standard RL (not risk-sensitive) approach and illustrates the difference due to the risk criteria and due to the difference in the learning strategies. Also, for the basic setting of CVaR, you can compare with existing approaches. I strongly recommend the authors to include an experiment section in their final submission.
> >
> > I will keep my score (7) based on the response by the authors.

---

> > > ### Author Response · Authors · 2022-08-09
> > > **Experiments**
> > >
> > > **Experiments.** Thanks for suggesting suitable baselines. We have run simulations on an instance of the standard frozen lake environment designed to have multiple paths with different risk-reward tradeoffs, using a CVaR objective. We have included some preliminary plots and discussion in Appendix E (highlighted in blue). At a high level, we find that (i) our approach converges, as expected, whereas a naive greedy exploration strategy as well as a naive strategy that uses UCBVI with the expected return objective both do not converge, and (ii) our approach performs worse as the CVaR parameter $\alpha$ becomes small, as expected according to our theory. These results validate our theoretical and algorithmic contributions.

---

### Official Review · Reviewer_yvdY · 2022-07-15

**Rating:** 5
**Confidence:** 3
**Soundness:** 3 good
**Presentation:** 3 good
**Contribution:** 3 good

**Summary:**

The paper studies the problem of risk-sensitive RL, and prove regret bound under a class of risk-sensitive objectives, including CVaR. The results build on a novel CVaR objective and optimistic MDP.

**Questions:**

- It would be nice to provide some explanations to explicitly connect the formulation here with existing settings.
- Also, while def. (1) appears to be more general, it would still help if the paper can discuss how it generalizes existing objectives.
- The reviewer would suggest adding a thorough comparison with existing works. In particular, how are the dependence on different parameters compared to existing results? The paper only mentions the dependence on K.
- The algorithmic aspect of the paper does not appear significant. In particular, the main results in the paper focus on showing equivalence or transformation. The proposed algorithm 1 does not appear to be novel (although this has nothing to do with its performance). The authors could perhaps better highlight the novelty of the results.


**Limitations:**

- The algorithmic part of the paper does not appear strong.
- A better comparison with recent results would be helpful.

**Strengths And Weaknesses:**

Strengths
- The problem of risk-sensitive RL is interesting and important.

Weaknesses
- The algorithmic part of the paper does not appear strong.
- A better comparison with recent results would be helpful.

---

> ### Author Response · Authors · 2022-08-02
> **Response to Reviewer yvdY**
>
> Thank you for your comments and feedback.
>
> **Objective.** We have provided some explanations with regard to the objective in the *Risk-sensitive objective* subsection in Section 2 (lines 63-67), and will expand this discussion. The objective we consider (originally studied in [1] for the RL setting) entails the usual expected cumulative reward objective by taking $G(\tau)=\text{Uniform}([0,1])$, where all the quantiles of the return are weighted equally. Choosing $G(\tau)=\tau\cdot\mathbb{1}(\tau\le\alpha)/\alpha$ for $\alpha\in[0,1]$ will result in the $\alpha$-conditional value at risk (CVaR) objective, where return is only averaged over the bottom $\alpha$-quantile. Finally, choosing $\alpha=1$ recovers the traditional expected return objective. Other risk sensitive-objectives can also be captured, for example the Wang measure [2], and the cumulative probability weighting (CPW) metric [3].
>
> **Comparison to previous results.** First, we note that our work proves the first regret bound for risk-sensitive reinforcement learning with a general class of risk-sensitive objectives. In particular, CVaR is the most important special case in the risk-sensitive objectives we consider, and no previous regret analysis for CVaR is known. Therefore, the closest existing bounds we may compare to are typical expected return objectives, which is a lower bound in our setting because expected return is a special case of CVaR.
> As mentioned in the paper, our algorithm achieves the optimal rate in the number of episodes $K$ in this sense. In terms of the dependence on the number of states $|\mathcal{S}|$, our bound has an extra $\sqrt{|\mathcal{S}|}$ factor compared to the UCRL2 algorithm [4], and an extra $|\mathcal{S}|$ factor compared to the improved bound of the UCBVI algorithm [5]. For the dependence on the number of actions $|\mathcal{A}|$, our bound has an extra $\sqrt{|\mathcal{S}|}$ factor compared to both UCRL2 and UCBVI. Our dependence on the horizon length $T$ is $T^2$, compared to a $T^{3/2}$ in UCBVI and $T$ in a variant of UCBVI [5] utilizing a carefully designed variance-based bonus. Finally, since the weighting function $G$ is only relevant in the form of the risk-sensitive objectives, there is no prior work to compare to in terms of the dependence on the Lipschitz constant $L_G$ of $G$. We will add this discussion to our paper.
>
> We believe that tightening dependence on these parameters is an important direction for future work, but the most important parameter remains $K$, for which we obtain optimal dependence. We note that for traditional RL with the expected return objective, the optimal dependences were only achieved after many papers spanning almost two decades.
>
> **Algorithmic contribution.** While our main contributions are theoretical, not algorithmic, our algorithm also has some novelty. In particular, we propose a novel optimistic exploration strategy tailored to the risk-sensitive setting. Specifically, we construct an optimistic MDP based on the empirical estimates by creating a fictitious “optimistic state” and inflating the reward probability measure. The standard UCBVI algorithm constructs an optimistic MDP by adding a bonus to the reward of a state-action pair that is inversely proportional to its uncertainty. However, we found that this strategy is not suitable for maximizing the risk-sensitive objective, which requires optimism over the return distribution. The extra state $s_\infty$ in our construction is crucial to dealing with this issue.
> We note that extended value iteration in the UCRL2 algorithm [4] also does not work here because we assume a blackbox oracle for a given MDP. In other words, we cannot search in a set of possible MDPs for the most optimistic one, but instead need to construct an explicit optimistic MDP for the oracle to compute an optimal policy.
>
> **References.**
>
> [1] Dabney et al., Implicit Quantile Networks for Distributional Reinforcement Learning. *International Conference on Machine Learning*, (2018).
>
> [2] Wang, A Class of Distortion Operators for Pricing Financial and Insurance Risks. *Journal of Risk and Insurance*, pages 15–36, 2000.
>
> [3] Tversky and Kahneman, Advances in Prospect Theory: Cumulative Representation of Uncertainty. *Journal of Risk and Uncertainty*, 5(4):297–323, 1992.
>
> [4] Auer et al., Near-Optimal Regret Bounds for Reinforcement Learning. *Advances in Neural Information Processing Systems*, 21, 2008.
>
> [5] Azar et al., Minimax regret bounds for reinforcement learning. *International Conference on Machine Learning*, pages 263-272. PMLR, 2017.

---

### Author Response · Authors · 2022-08-04
**Paper Revision**

We thank the reviewers for their helpful comments and feedback.

We have uploaded a revision of our paper according to the suggestions received. We mark the additions in blue, and list them below:

- A Related work subsection (current lines 46-55) at the end of Section 1: *Introduction*, covering related work on regret bounds for risk-sensitive RL.

- An expanded discussion of our risk-sensitive objective and its relation to existing risk measures in the subsection Risk-sensitive objective (current lines 75-81) in Section 2: *Problem Formulation*.

- A formal description of how to compute an optimal policy for the CVaR objective in Appendix D (referred to on current line 92).

- A comparison to previous theoretical guarantees in terms of other parameters $\mathcal{S}$, $\mathcal{A}$, and $T$ at the end of Section 4: *Upper Confidence Bound Algorithm* (current lines 173-181).

- A proof sketch for Theorem 4.1 at the beginning of Section 5: *Proof of Theorem 4.1* (current lines 184-199).

---

### Meta-Review · Area_Chair_CoyW · 2022-08-25

**Recommendation:** Accept
**Confidence:** Certain

**Metareview:**

This paper proposes an optimistic algorithm for regret minimization of risk-sensitive measures in tabular episodic MDPs. It shows that this algorithm achieves a $\sqrt{S^3 A^2 K} L poly(T)$ regret bound where $K$ is the number of episodes,  $T$ is the episode length and $L$ is the Lipschitz-constant of the weighting function associated with the risk measure (e.g. CVaR). This is the first regret bound in this setting.

The initial assessment by all reviewers was overall positive. They appreciated this first result in this new and important setting. However, there were also concerns, especially regarding the algorithmic contribution and the comparison to related work. The authors' response could largely address these issues. However, there are still some concerns regarding the tightness of the analysis and the relation to techniques in existing work. These are described in detail below and have been discussed in the reviewers, AC and SAC discussion.
All in all, the paper is recommended to be accepted because of its new guarantee in this new setting, motivating more research in this area that builds on this initial work. However, the authors are encouraged to further comment on the tightness of their analysis in the camera-ready version.

**Tightness of the Analysis**: The presented regret bound exhibits an additional $S \sqrt{A} L$ factor compared to state of the art regret bounds in the risk-neutral case. While a dependency on $L$ is expected in the risk-sensitive setting it is not clear that these additional $S$ and $A$ dependencies are necessary. The paper does not discuss where exactly these factors come from in the analysis and does not provide any regret lower bounds for this novel setting. While the rebuttal revision added a discussion to the paper on the existence of these additional factors compared to existing results for the risk-neutral case, several reviewers and the AC found this to be not quite satisfactory. The paper should discuss where these factors come from and why it would be difficult (if not impossible) to remove them.
Based on the AC's reading of the paper, these dependencies appear because of two places: (1) $\epsilon_P^{(k)}$ has a $\sqrt{S}$ dependency which seems to be derived from an $\ell_1$ concentration bound. However, it is not clear that this is really necessary and one could instead derive them from concentration arguments on individual $P(s'|s,a)$ probabilities since they are all down-shifted individually. (2) The final display in the proof of Thm 4.1 (Lines 475 and following) seemingly applies a standard argument in regret analyses but gets a worse dependency of S and A (linear instead of under the square root, compare for example to Eq 4.8 in https://arxiv.org/pdf/1807.03765.pdf).

**Award:**

No

---

### Decision · Program_Chairs · 2022-09-14

Accept